# Early-stopped neural networks are consistent

**Ziwei Ji**      **Justin D. Li**      **Matus Telgarsky**
<{ziweiji2,jdli3,mjt}@illinois.edu>
University of Illinois, Urbana-Champaign

## Abstract

This work studies the behavior of shallow ReLU networks trained with the logistic loss via gradient descent on binary classification data where the underlying data distribution is general, and the (optimal) Bayes risk is not necessarily zero. In this setting, it is shown that gradient descent with early stopping achieves population risk arbitrarily close to optimal in terms of not just logistic and misclassification losses, but also in terms of calibration, meaning the sigmoid mapping of its outputs approximates the true underlying conditional distribution arbitrarily finely. Moreover, the necessary iteration, sample, and architectural complexities of this analysis all scale naturally with a certain complexity measure of the true conditional model. Lastly, while it is not shown that early stopping is necessary, it is shown that any univariate classifier satisfying a *local interpolation property* is inconsistent.

## 1 Overview and main result

Deep networks trained with gradient descent seem to have no trouble adapting to arbitrary prediction problems, and are steadily displacing stalwart methods across many domains. In this work, we provide a mathematical basis for this good performance on arbitrary binary classification problems, considering the simplest possible networks: shallow ReLU networks where only the inner (input-facing) weights are trained via vanilla gradient descent with a constant step size. The central contributions are as follows.

1. **Fully general classification tasks.** The joint distribution generating the $(x, y)$ pairs only requires $x$ to be bounded, and is otherwise arbitrary. In particular, the underlying distribution may be noisy, meaning the true conditional model of the labels, $\Pr[Y = 1 | X = x]$, is arbitrary.

    In this setting, we show that as data, width, and training time increase, the logistic loss *measured over the population* converges to optimality over all measurable functions, which moreover implies that the induced conditional model (defined by a sigmoid mapping) converges to the true model, and the population misclassification rate also converges to optimality. This is in contrast with prior analyses of gradient descent, which either only consider the training risk [Allen-Zhu et al., 2018b, Du et al., 2019, Zou et al., 2018, Oymak and Soltanolkotabi, 2019, Song and Yang, 2019], or can only handle restricted conditional models [Allen-Zhu et al., 2018a, Arora et al., 2019, Cao and Gu, 2019, Nitanda and Suzuki, 2019, Ji and Telgarsky, 2020b, Chen et al., 2021].

2. **Adaptivity to data simplicity.** The required number of data samples, network nodes, and gradient descent iterations all shrink if the *distribution* satisfies a natural notion of simplicity: the true conditional model $\Pr[Y = 1 | X = x]$ is approximated well by a low-complexity infinite-width random feature model.

Rounding out the story and contributions, firstly we present a brief toy univariate model hinting towards the necessity of early stopping: concretely, any univariate predictor satisfying a *local interpolation property* can not achieve optimal test error for noisy distributions. Secondly, our analysis is backed by a number of lemmas that could be useful elsewhere; amongst these are a

35th Conference on Neural Information Processing Systems (NeurIPS 2021).

*multiplicative error* property of the logistic loss, and separately a technique to control the effects of large network width over not just a finite sample, but over the entire sphere.

## 1.1 Main result: optimal test error via gradient descent

The goal in this work is to minimize the logistic risk over the population: letting $\mu$ denote an arbitrary Borel measure over $(x, y)$ pairs with compactly-supported marginal $\mu_x$ and conditional $p_y$, with a data sample $((x_i, y_i))_{i=1}^n$, and a function $f$, define the logistic loss, empirical logistic risk, and logistic risk respectively as

$$\ell(r) := \ln(1 + e^{-r}), \qquad \widehat{\mathcal{R}}(f) := \frac{1}{n} \sum_{k=1}^n \ell(y_k f(x_k)), \qquad \mathcal{R}(f) := \mathbb{E}_{x,y} \ell(y f(x)).$$

We use the logistic loss not only due to its practical prevalence, but also due to an interesting multiplicative error property which strengthens our main results (cf. Lemma B.1 and Theorem 1.1), all while being Lipschitz.

We seek to make the risk $\mathcal{R}(f)$ as small as possible: formally, we compare against the Bayes risk

$$\overline{\mathcal{R}} := \inf \left\{ \mathcal{R}(f) \ : \ \text{measurable } f : \mathbb{R}^d \to \mathbb{R} \right\}.$$

While competing with $\overline{\mathcal{R}}$ may seem a strenuous goal, in fact it simplifies many aspects of the learning task. Firstly, due to the universal approximation properties of neural networks [Funahashi, 1989, Hornik et al., 1989, Cybenko, 1989, Barron, 1993], we are effectively working over the space of all measurable functions already. Secondly, as will be highlighted in the main result below, via the theory of classification calibration [Zhang, 2004, Bartlett et al., 2006], competing with the Bayes (convex) risk also recovers the true conditional model, and minimizes the misclassification loss; this stands in contrast with the ostensibly more modest goal of minimizing misclassification over a restricted class of predictors, namely the *agnostic learning* setting, which suffers a variety of computational and statistical obstructions [Goel et al., 2020a,b, Yehudai and Shamir, 2020, Frei et al., 2020].

Our predictors are shallow ReLU networks, trained via gradient descent — the simplest architecture which is not convex in its parameters, but satisfies universal approximation. In detail, letting $(a_j)_{j=1}^m$ be uniformly random $\pm 1$ signs, $(w_j)_{j=1}^m$ with $w_j \in \mathbb{R}^d$ be standard Gaussians, and $\rho > 0$ be a *temperature*, we predict on an input $x \in \mathbb{R}^d$ with

$$f(x; \rho, a, W) := f(x; W) := \frac{\rho}{\sqrt{m}} \sum_{j=1}^m a_j \sigma_{\mathrm{r}}(w_j^\mathsf{T} x),$$

where $\sigma_{\mathrm{r}}(z) := \max\{0, z\}$ is the ReLU; since only $W$ is trained, both $\rho$ and $a$ are often dropped. To train, we perform gradient descent with a constant step size on the empirical risk:

$$W_{i+1} := W_i - \eta \nabla \widehat{\mathcal{R}}(W_i), \qquad \text{where } \widehat{\mathcal{R}}(W) := \widehat{\mathcal{R}}\left(x \mapsto f(x; W)\right).$$

Our guarantees are for an iterate with small empirical risk and small norm: $W_{\leq t} := \arg\min\{\widehat{\mathcal{R}}(W_i) : i \leq t, \|W_i - W_0\| \leq R_{\mathrm{gd}}\}$, where $R_{\mathrm{gd}}$ is our *early stopping radius*: if $R_{\mathrm{gd}}$ is guessed correctly, our rates improve, but our analysis also handles the case $R_{\mathrm{gd}} = \infty$ where no guess is made, and indeed this is used in our final consistency analysis (a pessimistic, fully general setting).

Our goal is to show that this iterate $W_{\leq t}$ has approximately optimal *population* risk: $\mathcal{R}(W_{\leq t}) \approx \overline{\mathcal{R}}$. Certain prediction problems may seem simpler than others, and we want our analysis to reflect this while abstracting away as many coincidences of the training process as possible. Concretely, we measure simplicity via the performance and complexity of an infinite-width random feature model over the true distribution, primarily based on the following considerations.

- By measuring performance over the population, random effects of the training sample are removed, and it is impossible for the random feature model to simply revert to memorizing data, as it never sees that training data.
- The random feature model has infinite width, and via sampling can be used as a benchmark for all possible widths simultaneously, but is itself freed from coincidences of random weights.

In detail, our infinite-width random feature model is as follows. Let $\overline{U}_\infty : \mathbb{R}^d \to \mathbb{R}^d$ be an (uncountable) collection of weights (indexed by $\mathbb{R}^d$), and define a prediction mapping via

$$f(x; \overline{U}_\infty) := \int \left\langle \overline{U}_\infty(v), x\mathbb{1}[v^\mathsf{T} x \geq 0] \right\rangle \mathrm{d}\mathcal{N}(v), \qquad \text{whereby } \mathcal{R}(\overline{U}_\infty) := \mathcal{R}(x \mapsto f(x; \overline{U}_\infty)).$$

Note that for each Gaussian random vector $v \sim \mathcal{N}$, we construct a *random feature* $x \mapsto x\mathbb{1}[v^\mathsf{T} x \geq 0]$. This particular choice is simply the gradient of a corresponding ReLU $\nabla_v \sigma_\mathrm{r}(v^\mathsf{T} x)$, and is motivated by the NTK literature [Jacot et al., 2018, Li and Liang, 2018, Du et al., 2019]. A similar object has appeared before in NTK convergence analyses [Nitanda and Suzuki, 2019, Ji and Telgarsky, 2020b], but the conditions on $\overline{U}_\infty$ were always strong (e.g., data separation with a margin).

What, then, does it mean for the data to be simple? In this work, it is when there exists a $\overline{U}_\infty$ with $\mathcal{R}(\overline{U}_\infty) \approx \overline{\mathcal{R}}$, and moreover $\overline{U}_\infty$ has low norm; for technical convenience, we measure the norm as the maximum over individual weight norms, meaning $\sup_v \|\overline{U}_\infty(v)\|$. To measure approximability, for sake of interpretation, we use the *binary Kullback-Leibler divergence (KL)*: defining a conditional probability model $\phi_\infty$ corresponding to $\overline{U}_\infty$ via

$$\phi_\infty(x) := \phi(f(x; \overline{U}_\infty)), \qquad \text{where } \phi(r) := \frac{1}{1 + \exp(-r)},$$

then the binary KL can be written as

$$\mathcal{K}_\mathrm{bin}(p_y, \phi_\infty) := \int \left( p_y \ln \frac{p_y}{\phi_\infty} + (1 - p_y) \ln \frac{1 - p_y}{1 - \phi_\infty} \right) \mathrm{d}\mu_x = \mathcal{R}(\overline{U}_\infty) - \overline{\mathcal{R}}.$$

This relationship between binary KL and the excess risk is a convenient property of the logistic loss, which immediately implies calibration as a consequence of achieving the optimal risk.

The pieces are all in place to state our main result.

**Theorem 1.1.** *Let width $m \geq \ln(emd)$, temperature $\rho > 0$, and reference model $\overline{U}_\infty$ be given with $R := \max\{4, \rho, \sup_v \|\overline{U}_\infty(v)\|\} < \infty$, and define a corresponding conditional model $\phi_\infty(x) := \phi(f(x; \overline{U}_\infty))$. Let optimization accuracy $\epsilon_\mathrm{gd}$ and radius $R_\mathrm{gd} \geq R/\rho$ be given, define effective radius $B := \min \left\{ R_\mathrm{gd}, \frac{3R}{\rho} + \frac{4e}{\rho} \sqrt{t} \sqrt{e^{\tau_0} \mathcal{R}(\overline{U}_\infty) + R\tau_n} \right\}$, and generalization, linearization, and sampling errors $(\tau_n, \tau_1, \tau_0)$ as*

$$\tau_n := \widetilde{\mathcal{O}} \left( \frac{(d \ln(1/\delta))^{3/2}}{\sqrt{n}} \right), \; \tau_1 := \widetilde{\mathcal{O}} \left( \frac{\rho B^{4/3} \sqrt{d \ln(1/\delta)}}{m^{1/6}} \right), \; \tau_0 := \widetilde{\mathcal{O}} \left( \rho \ln(1/\delta) + \frac{\sqrt{d \ln(1/\delta)}}{m^{1/4}} \right),$$

*where it is assumed $\tau_1 \leq 2$, and $\widetilde{\mathcal{O}}$ hides constants and $\ln(nmd)$. Choose step size $\eta := 4/\rho^2$, and run gradient descent for $t := 1/(8\epsilon_\mathrm{gd})$ iterations, selecting iterate $W_{\leq t} := \arg\min\{\widehat{\mathcal{R}}(W_i) : i \leq t, \|W_i - W_0\| \leq R_\mathrm{gd}\}$. Then, with probability at least $1 - 25\delta$,*

$$
\begin{aligned}
& \mathcal{R}(W_{\leq t}) - \overline{\mathcal{R}} && \textit{(logistic error)} \\
\leq \quad & \mathcal{K}_\mathrm{bin}(p_y, \phi_\infty) + \left( e^{\tau_1 + \tau_0} - 1 \right) \mathcal{R}(\overline{U}_\infty) && \textit{(reference model error)} \\
+ \quad & e^{\tau_1} R^2 \epsilon_\mathrm{gd} && \textit{(optimization error)} \\
+ \quad & e^{\tau_1} (\rho B + R)\tau_n && \textit{(generalization error),}
\end{aligned}
$$

*where the classification and calibration errors satisfy*

$$
\begin{aligned}
& \mathcal{R}(W_{\leq t}) - \overline{\mathcal{R}} && \textit{(logistic error)} \\
\geq \quad & 2 \int \left( \phi(f(x; W_{\leq t})) - p_y \right)^2 \mathrm{d}\mu_x(x) && \textit{(calibration error)} \\
\geq \quad & \frac{1}{2} \left( \mathcal{R}_\mathrm{z}(W_{\leq t}) - \overline{\mathcal{R}}_\mathrm{z} \right)^2 && \textit{(classification error).}
\end{aligned}
$$

*Lastly, for any $\epsilon > 0$, there exists $\overline{U}_\infty^{(\epsilon)}$ with $\sup_v \|\overline{U}_\infty^{(\epsilon)}(v)\| < \infty$ and whose conditional model $\phi_\infty^{(\epsilon)}(x) := \phi(f((x, 1)/\sqrt{2}; \overline{U}_\infty^{(\epsilon)}))$ satisfies $\mathcal{K}_\mathrm{bin}(p_y, \phi_\infty^{(\epsilon)}) \leq \epsilon$.*

**Remark 1.1.** The key properties of Theorem 1.1 are as follows.

1. **(Achieving error $\mathcal{O}(\epsilon)$ in three different regimes.)** As Theorem 1.1 is quite complicated, consider three different situations, which vary the reference model $\overline{U}_\infty$ and its norm upper bound $R := \max\{4, \rho, \sup_v \|\overline{U}_\infty(v)\|\} < \infty$, as well as the early stopping radius $R_{\mathrm{gd}}$. Let target population (excess) risk $\epsilon > 0$ be given, set $\epsilon_{\mathrm{gd}} = \epsilon$ and $t = 1/(8\epsilon_{\mathrm{gd}})$ as in Theorem 1.1, and suppose $n \geq 1/\epsilon^2$ samples: in each of the three following settings, the other parameters parameters (namely $\rho$ and $m$) will be chosen to ensure a final error $\mathcal{R}(W_{\leq t}) - \overline{\mathcal{R}} = \mathcal{O}(\epsilon)$.

    (a) **(Easy data.)** Suppose a setting with *easy data*: specifically, suppose that for chosen target accuracy $\epsilon > 0$, there exists $\overline{U}_\infty$ with $\mathcal{K}_{\mathrm{bin}}(p_y, \phi_\infty) = \mathcal{R}(\overline{U}_\infty) - \overline{\mathcal{R}} \leq \mathcal{R}(\overline{U}_\infty) \leq \epsilon$. If we set $\rho = 1$ and $m \geq R^8$, then $(\tau_n, \tau_1, \tau_0)$ are all constant, and we get a final bound $\mathcal{R}(W_{\leq t}) - \overline{\mathcal{R}} = \mathcal{O}(\epsilon)$.
    Note crucially that $m \approx R^8$ sufficed for this setting; this was a goal of the present analysis, as it recovers the *polylogarithmic width* analyses from prior work [Ji and Telgarsky, 2020b, Chen et al., 2021]. Those works however either used a separation condition due to Nitanda and Suzuki [2019] in the shallow case, or an assumption on the approximation properties of the sampled weights (a random variable) in the deep case, and thus the present analysis provides not just a re-proof, but a simplification and generalization. This was the motivation for the strange *multiplicative* form of the errors in Theorem 1.1: had we used the more common additive errors with standard linearization tools, a polylogarithmic width proof would fail.

    (b) **(General data, clairvoyant early stopping radius $R_{\mathrm{gd}}$.)** Suppose that we are in the general noisy case, meaning any $\overline{U}_\infty$ we pick has a large error $\mathcal{K}_{\mathrm{bin}}(p_y, \phi_\infty)$, but we magically know the $R$ corresponding to a good $\overline{U}_\infty$, and can choose $R_{\mathrm{gd}} = R/\rho$. Unlike the previous case, to achieve some target error $\epsilon$, we need to work harder to control the term $\left[\exp(\tau_1 + \tau_0) - 1\right] \mathcal{R}(\overline{U}_\infty)$, since we no longer have small $\mathcal{R}(\overline{U}_\infty)$; to this end, since $\tau_1 = \widetilde{\mathcal{O}}(R^{4/3}/(m\rho^2)^{1/6})$ and $\tau_0 = \widetilde{\mathcal{O}}(\rho + 1/m^{1/4})$, choosing $\rho = m^{-1/8}$ and $m = 1/\epsilon^8$ gives $\tau_1 = \widetilde{\mathcal{O}}(\epsilon)$ and $\tau_0 = \widetilde{\mathcal{O}}(\epsilon)$, and together $\mathcal{R}(W_{\leq t}) - \overline{\mathcal{R}} = \mathcal{O}(\epsilon)$.

    (c) **(General data, worst-case early stopping.)** Suppose again the case of general noisy data with large error $\mathcal{K}_{\mathrm{bin}}(p_y, \phi_\infty)$ for any $\overline{U}_\infty$ we pick, but now suppose we have no early stopping hint, and pessimistically set $R_{\mathrm{gd}} = \infty$. As a consequence of all of this, the term $B$ can scale as $t^{2/3}/\rho = 1/(\rho\epsilon^{2/3})$, thus to control $\tau_1 = \widetilde{\mathcal{O}}((1/\epsilon)^{2/3}/(m\rho^2)^{1/6})$ and $\tau_0 = \widetilde{\mathcal{O}}(\rho + 1/m^{1/4})$, we can again choose $\rho = m^{-1/8}$, but need a larger width $m = 1/\epsilon^{40/3}$. Together, we once again achieve population excess risk $\mathcal{R}(W_{\leq t}) - \overline{\mathcal{R}} = \mathcal{O}(\epsilon)$.

    Summarizing, a first key point is that arbitrarily small excess risk $\mathcal{O}(\epsilon)$ is always possible; as discussed, this is in contrast to prior work, which either only gave training error guarantees, or required restrictive conditions for small test error. A second key point is that the parameters of the bound, most notably the required width, will shrink greatly when either the data is easy, or an optimal stopping radius $R_{\mathrm{gd}}$ is known.

2. **(Consistency.)** *Consistency* is a classical statistical goal of achieving the optimal test error almost surely *over all possible predictors* as $n \to \infty$; here it is proved as a consequence of Theorem 1.1, namely the preceding argument that we can achieve excess risk $\mathcal{O}(\epsilon)$ even with general prediction problems and no early stopping hints ($R_{\mathrm{gd}} = \infty$). The consistency guarantee is stated formally in Corollary 2.3. The statement takes the width to infinity, and demonstrates another advantage of using an infinite-width reference model: within the proof, after fixing a target accuracy, the reference model is *fixed* and used for all widths simultaneously.

3. **(Non-vacuous generalization, and an estimate of $R$.)** There is extensive concern throughout the community that generalization estimates are hopelessly loose [Neyshabur et al., 2014, Zhang et al., 2016, Dziugaite and Roy, 2017]; to reduce the concern here, we raise two points. Firstly, these concerns usually involve explicit calculations of generalization bounds which have terms scaling with some combination of $\|W\|$ (not $\|W - W_0\|$) and $m$; e.g,. one standard bound has spectral norms $\|W\|_2$ and $(2, 1)$ matrix norms $\|(W - W_0)^\top\|_{2,1}$, which are upper bounded by $\|W - W_0\|\sqrt{m}$ [Bartlett et al., 2017]. By contrast, the present work uses a new generalization

bound technique (cf. Lemma B.8) which first *de-linearizes* the network, then applies a *linear generalization bound* which has only $\|W - W_0\|$ and no explicit $\mathrm{poly}(m)$, and then *re-linearizes*.

Secondly, there may still be concern that the story here is broken due to the term $R$, and namely the non-existence of good choices for $\overline{U}_\infty$. For this, we conducted a simple experiment. Noting that we can freeze the initial features and train linear predictors of the form $f^{(0)}(x; V)$ for weights $V \in \mathbb{R}^{m \times d}$ (cf. section 1.4), and that the performance converges to the infinite-width performance as $m \to \infty$, we fixed a large width and trained two prediction tasks: an *easy* task of MNIST 1 vs 5 until $R_{\text{easy}}/\sqrt{n} \approx 1/2$, and a *hard* task of MNIST 3 vs 5 until $R_{\text{hard}}/\sqrt{n} \approx 1/2$. After training, we obtained test error $\mathcal{R}(V_{\text{easy}}) \approx 0.01$ and $\mathcal{R}(V_{\text{hard}}) \approx 0.08$. Plugging all of these terms back in to the bound, firstly these techniques can yield a non-vacuous generalization bound, secondly they do not exhibit bad scaling with large width, and thirdly they do reflect the difficulty of the problem, as desired.

4. **(Early stopping and the NTK.)** As discussed above, when the data is noisy, the method is explicitly early stopped, either by clairvoyantly choosing $R_{\text{gd}}$, or by making $t$ small. In this setting, the optimization accuracy $\epsilon_{\text{gd}}$ is an *excess* empirical risk, meaning in particular that 0 training error (the *interpolation regime* [Belkin et al., 2018a]) will *not* be reached. This is in stark contrast to standard NTK analyses [Allen-Zhu et al., 2018b], which guarantee zero training error, but can not ensure good test error in general. Since the NTK itself is an early stopping (as in, if one continues to optimizes, one exits the NTK), then the early stopping in this work is even earlier than the NTK early stopping; this situation is summarized in Figure 1 in the appendix, and will be revisited for the lower bound in Section 1.2.

5. **(Classification and calibration.)** The relationship to classification and calibration errors is merely a restatement of existing results [Zhang, 2004, Bartlett et al., 2006], though it is reproved here in an elementary way for the special case of the logistic loss. Similarly, the guarantee that $\mathcal{K}_{\text{bin}}(p_y, \phi_\infty^{(\epsilon)})$ can be made arbitrarily small is also not a primary contribution, and indeed most of the heavy lifting is provided both by prior work in neural network approximation [Barron, 1993], and by the existing and reliable machinery for proving consistency [Schapire and Freund, 2012]. As such, the consistency result is stated only much later in Corollary 2.3, and our focus is on the exact risk guarantees in Theorem 1.1.

6. **(Inputs with bias:** $(x, 1)/\sqrt{2} \in \mathbb{R}^{d+1}$**.)** The end of Theorem 1.1 appends a constant to the input (and rescales), which simulates a bias term inside each ReLU; this is necessary since our models are (sigmoid mappings of) homogeneous functions, whereas $p_y$ is general. Biases are also simulated in this way in the consistency result in Corollary 2.3.

Further discussion of Theorem 1.1, including the formal consistency result (cf. Corollary 2.3) and a proof sketch, all appear in Section 2. Full proofs appear in the appendices.

## 1.2 Should we early stop?

Theorem 1.1 uses early stopping: it can blow up if $\overline{\mathcal{R}} > 0$ and the two gradient descent parameters $R_{\text{gd}}$ and $1/\epsilon_{\text{gd}}$ are taken to $\infty$ in an uncoordinated fashion. Part of this is purely technical: as with many neural network optimization proofs, the analysis breaks when far from initialization. It is of course natural to wonder what happens if one trains indefinitely, entering the actively-studied *interpolation regime* [Belkin et al., 2018b,a, Bartlett et al., 2019]. Furthermore, there is evidence that gradient descent on shallow networks limits towards a particular interpolating choice, one with large margins [Soudry et al., 2018, Lyu and Li, 2020, Chizat and Bach, 2020, Ji and Telgarsky, 2020a]. Is this behavior favorable?

While we do not rule out that the interpolating solutions found by neural networks perform well, we show that at least in the low-dimensional (univariate!) setting, if a prediction rule perfectly labels the data and is not too wild between training points, then it is guaranteed to achieve poor test loss on noisy problems. This negative observation is not completely at odds with the interpolation literature, where the performance of some rules improves with dimension [Belkin et al., 2018b].

**Proposition 1.2.** *Given a finite sample $((x_i, y_i))_{i=1}^n$ with $x_i \in \mathbb{R}$ and $y_i \in \{\pm 1\}$, let $\mathcal{F}_n$ denote the collection of* local interpolation rules *(cf. Figure 2 in the appendix): letting $x_{(i)}$ index examples in*

*sorted order, meaning $x_{(1)} \leq x_{(2)} \leq \cdots \leq x_{(n)}$, define $\mathcal{F}_n$ as*

$$\mathcal{F}_n := \big\{ f : \mathbb{R} \to \mathbb{R} \; : \; \forall i \; f(x_{(i)}) = y_{(i)}, \; and$$
$$if \; y_{(i)} = y_{(i+1)}, \; then \; \inf_{\alpha \in [0,1]} f\big(\alpha x_{(i)} + (1-\alpha)x_{(i+1)}\big) y_{(i)} > 0 \big\}.$$

*Then there exists a constant $c > 0$ so that with probability at least $1 - \delta$ over the draw of $((x_i, y_i))_{i=1}^n$ with $n \geq \ln(1/\delta)/c$, every $f \in \mathcal{F}_n$ satisfies $\mathcal{R}_z(f) \geq \check{\mathcal{R}}_z(f) + c$.*

Although a minor contribution, this result will be discussed briefly in Section 3, with detailed proofs appearing in the appendices. For a similar discussion for nearest neighbor classifiers albeit under a few additional assumptions, see [Nakkiran and Bansal, 2021].

### 1.3 Related work

**Analyses of gradient descent.** The proof here shares the most elements with recent works whose width could be polylogarithmic in the sample size and desired target accuracy $1/\epsilon$ [Ji and Telgarsky, 2020b, Chen et al., 2021]. Similarities include using a regret inequality as the core of the proof, using an infinite-width target network [Nitanda and Suzuki, 2019, Ji and Telgarsky, 2020b], and using a *linearization inequality* [Chen et al., 2021, Allen-Zhu et al., 2018b]. On the technical side, the present work differs in the detailed treatment of the logistic loss, and in the linearization inequality which is extended to hold over the population risk; otherwise, the core gradient descent analysis here is arguably simplified relative to these prior works. It should be noted that the use of a regret inequality here and in the previous works crucially makes use of a negated term which was dropped in some classical treatments; this trick is now re-appearing in many places [Orabona and Pál, 2021, Frei et al., 2020].

There are many other, somewhat less similar works in the vast literature of gradient descent on neural networks, in particular in the neural tangent regime [Jacot et al., 2018, Li and Liang, 2018, Du et al., 2019]. These works often handle not only training error, but also testing error [Allen-Zhu et al., 2018a, Arora et al., 2019, Cao and Gu, 2019, Nitanda and Suzuki, 2019, Ji and Telgarsky, 2020b, Chen et al., 2021]. As was mentioned before, these works do not appear to handle arbitrary target models; see for instance the modeling discussion in [Arora et al., 2019, Section 6]. As another interesting recent example, some works explicitly handle certain noisy conditional models, but with error terms that do not go to zero in general [Liang et al., 2021].

**Consistency.** Consistency of deep networks with classification loss and *some* training procedure is classical; e.g., in [Farago and Lugosi, 1993], the authors show that it suffices to run a computationally intractable algorithm on an architecture chosen to balance VC dimension and universal approximation. Similarly, the work here makes use of Barron's superposition analysis in an infinite-width form to meet the Bayes risk [Barron, 1993, Ji et al., 2020b]. The statistics literature has many other works giving beautiful analyses of neural networks, e.g., even with minimax rates [Schmidt-Hieber, 2017], though it appears this literature generally does not consider gradient descent and arbitrary classification objectives.

In the boosting literature, most consistency proofs only consider classification loss [Bartlett and Traskin, 2007, Schapire and Freund, 2012], though there is a notable exception which controls the convex loss (and thus calibration), although the algorithm has a number of modifications [Zhang and Yu, 2005]. In all these works, arbitrary $p_y$ are not handled explicitly as here, but rather *implicitly* via assumptions on the expressiveness of the weak learners. One exception is the logistic loss boosting proof of Telgarsky [2013], which explicitly handles measurable $p_y$ via Lusin's theorem as is done here, but ultimately the proof only controls classification loss.

Following the arXiv posting of this work, a few closely related works appeared. Firstly, Richards and Kuzborskij [2021] show that the expected excess risk can scale with $\|W_t - W_0\|_F/n^\alpha$, though in contrast with the present work, it is not shown that this ratio can go to zero for arbitrary prediction problems, and moreover the bound is in expectation only. Secondly, the work of Braun et al. [2021] is even closer, however it requires a condition on the Fourier spectrum of the conditional model $p_y$, which is circumvented here via a more careful Fourier analysis due to Ji et al. [2020b].

**Calibration.** There is an increasing body of work considering the (in)ability of networks trained with the logistic loss to recover the underlying conditional model. Both on the empirical side [Guo

et al., 2017] and on the theoretical side [Bai et al., 2021], the evidence is on the side of the logistic loss doing poorly, specifically being *overconfident*, meaning the sigmoid outputs are too close to 0 or 1. This overconfident regime corresponds to large margins; indeed, since gradient descent can be proved in some settings to exhibit unboundedly large unnormalized margins on all training points [Lyu and Li, 2020], the sigmoid mapping of the predictions will necessarily limit to exactly 0 or 1. On the other hand, as mentioned in [Bai et al., 2021], regularization suffices to circumvent this issue. In the present work, a combination of early stopping and small temperature are employed. As mentioned before, calibration is proved here as an immediate corollary of meeting the optimal logistic risk via classification calibration [Zhang, 2004, Bartlett et al., 2006].

### 1.4 Further notation and technical background

The loss $\ell$, risks $\mathcal{R}$ and $\widehat{\mathcal{R}}$, and network $f$ have been defined. The misclassification risk $\mathcal{R}_z(f) = \Pr[\text{sgn}(f(X)) \neq Y]$ appeared in Theorem 1.1, where $\text{sgn}(f(x)) = 2 \cdot \mathbb{1}[f(x) \geq 0] - 1$.

Next, consider the "gradient" of $f$ with respect to weights $W$:

$$\nabla f(x; W) := \frac{\rho}{\sqrt{m}} \sum_{j=1}^{m} a_j \mathbb{1}[w_j^\mathsf{T} x \geq 0] e_j x^\mathsf{T};$$

it may seem the nondifferentiability at 0 is concerning, but in analyses close to initialization (as is the one here), few activations change, and their behavior is treated in a worst-case fashion. Note that, as is easily checked with this expression, $\|\nabla f(W)\| \leq \rho$, which is convenient in many places in the proofs. Here $\|\cdot\|$ denotes the Frobenius norm; $\|\cdot\|_2$ will denote the spectral norm.

Given weight matrix $W_i$ at time $i$, let $(w_{i,j}^\mathsf{T})_{j=1}^m$ refer to its rows. Define features $f^{(i)}$ at time $i$ and a corresponding empirical risk $\widehat{\mathcal{R}}^{(i)}$ using the features at time $i$ as

$$f^{(i)}(x; V) := \left\langle \nabla f(x; W_i), V \right\rangle = \frac{\rho}{\sqrt{m}} \sum_j a_j v_j^\mathsf{T} x \mathbb{1}[w_{i,j}^\mathsf{T} x \geq 0],$$

$$\widehat{\mathcal{R}}^{(i)}(x; V) := \widehat{\mathcal{R}}(x \mapsto f^{(i)}(x; V)).$$

By 1-homogeneity of the ReLU, $f^{(i)}(x; W_i) = f(x; W_i)$, which will also be used often. These features at time $i$, meaning $f^{(i)}$ and $\widehat{\mathcal{R}}^{(i)}$, are very useful in analyses near initialization, as they do not change much. As such, $f^{(0)}$ and $\mathcal{R}^{(0)}$ and $\widehat{\mathcal{R}}^{(0)}$ will all appear often as well.

To be a bit pedantic about the measure $\mu$: as before, there is a joint distribution $\mu$, which is over the Borel $\sigma$-algebra on $\mathbb{R}^d \times \{\pm 1\}$, where $\|x\| \leq 1$ almost surely. This condition suffices to grant both a *disintegration* of $\mu$ into marginal $\mu_x$ and conditional $p_y$ [Kallenberg, 2002, Chapter 6], and also Lusin's theorem [Folland, 1999, Theorem 7.10], which is used to switch from a measurable function to a continuous one in the consistency proof (cf. Corollary 2.3).

## 2 Discussion and proof sketch of Theorem 1.1

This section breaks down the proof and discussion into four subsections: a section with common technical tools, then sections for the analysis of generalization, optimization, and approximation.

### 2.1 Key technical lemmas

There are two main new technical ideas which power many parts of the proofs: a multiplicative error property of the logistic loss, and a *linearization over the sphere*.

The logistic loss property is simple enough: for any $a \geq b$, it holds that $\ell(-a)/\ell(-b) \leq \exp(a - b)$. On the surface, this seems innocuous, but this simple inequality allows us to reprove existing polylogarithmic width results for easy data [Ji and Telgarsky, 2020b, Chen et al., 2021], however making use of a proof scheme which is slightly more standard, or at the very least more apparently a smooth convex proof with just this one special property of the logistic loss (as opposed to a few special properties).

The second tool is more technical, and is used crucially in many places in the proof. Many prior analyses near initialization bound the quantity

$$f(x; V) - f(x; W) - \langle \nabla f(x; W), V - W \rangle,$$

where $V$ and $W$ are both close to initialization [Allen-Zhu et al., 2018b, Cao and Gu, 2019, Chen et al., 2021]. These proofs are typically performed on a fixed example $x_k$, and then a union bound carries them over to the whole training set. Here, instead, such a bound is extended to hold *over the entire sphere*, as follows.

**Lemma 2.1** (Simplification of Lemma B.7). *Let scalars $\delta > 0$ and $R_V \geq 1$ and $R_B \geq 0$ be given.*

1. *With probability at least $1 - 3n\delta$,*

$$\sup_{\substack{\|W_i - W_0\| \leq R_V \\ \|W_j - W_0\| \leq R_V \\ \|B - W_0\| \leq R_B}} \frac{\widehat{\mathcal{R}}^{(i)}(B)}{\widehat{\mathcal{R}}^{(j)}(B)} \leq \exp\left(\frac{6\rho\left(R_B + 2R_V\right)R_V^{1/3}\ln(e/\delta)^{1/4}}{m^{1/6}}\right).$$

2. *Suppose $m \geq \ln(edm)$. With probability at least $1 - (1 + 3(d^2 m)^d)\delta$,*

$$\sup_{\|W_i - W_0\| \leq R_V} \frac{\mathcal{R}(W_i)}{\mathcal{R}^{(0)}(W_i)} \leq \exp\left(\frac{25\rho R_V^{4/3}\sqrt{\ln(edm/\delta)}}{m^{1/6}}\right).$$

The preceding lemma combines both the linearization technique and the multiplicative error property: it bounds how much the empirical and true risk change for a fix weight matrix if we swap in and out the features at different iterations. That these bounds are a ratio is due to the multiplicative error property. That the second part holds over the true risk, in particular controlling behavior over all $\|x\| \leq 1$, is a consequence of the new more powerful linearization technique. This linearization over the sphere is used crucially in three separate places: we use it when controlling the range in the generalization proofs, when *de-linearizing* after generalization, and when sampling from the infinite-width model $\overline{U}_\infty$. The method of proof is inspired by the concept of *co-VC dimension* [Gurvits and Koiran, 1995]: the desired inequality is first union bounded over a cover of the sphere, and then relaxed to all points on the sphere. A key difficulty here is the non-smoothness of the ReLU, and a key lemma establishes a smoothness-like inequality (cf. Lemma B.5). These techniques appear in full in the appendices.

## 2.2 Generalization analysis

The generalization analysis ends up being easy thanks to the multiplicative error control in Lemma B.1, and the aforementioned linearization over all points in the sphere. Specifically, to prove generalization, the network is first linearized, then generalization *of linear predictors* is applied, and then de-linearization is applied on the population risk side. This generalization bound only pays logarithmically in the width $m$.

Typically the easiest step in proving generalization is to provide a worst-case estimate on the range of the predictor. Here, since there is a goal of controlling the logistic loss over the population, brute forcing this range estimate incurs a polynomial dependence on network width. The solution here is again to apply the aforementioned Lemma B.3; the generalization statement appears in full as Lemma B.8 together with its proof in the appendices.

## 2.3 Gradient descent analysis

A common tool in linear prediction is the regret inequality

$$\|v_t - z\|^2 + 2\eta \sum_{i<t} \widehat{\mathcal{R}}(v_{i+1}) \leq \|v_0 - z\|^2 + 2t\eta\widehat{\mathcal{R}}(z),$$

which can be derived by expanding the square in $\|v_t - z\|^2$ and applying smoothness and convexity. The term $\|v_t - z\|^2$ is often dropped, but can be used in a very convenient way: by the triangle inequality, if $\|v_t - v_0\| \geq 2\|z - v_0\|$, then the norm terms above may be canceled from both sides,

which leaves only the empirical risk terms; overall, this argument ensures both small norm and small empirical risk. This idea has appeared in a variety of works [Shamir, 2020, Ji et al., 2020a], and is used here to provide a convenient norm control, allowing linearization and all other proof parts to go through. Combining this idea with the earlier generalization analysis and a few other minor tricks gives the following bounds, which in turn provide most of Theorem 1.1.

**Lemma 2.2.** *Let temperature $\rho > 0$, step size $\eta \leq 4/\rho^2$, optimization accuracy $\epsilon_{\mathrm{gd}} > 0$, radius $R_{\mathrm{gd}} > 0$, network width $m \geq \ln(emd)$, reference matrix $Z \in \mathbb{R}^{m \times d}$, corresponding scalar $R_Z \leq R_{\mathrm{gd}}$ where $R_Z \geq \max\{1, \eta\rho, \|W_0 - Z\|\}$, and $t \geq 1/(2\eta\rho^2\epsilon_{\mathrm{gd}})$ be given; correspondingly define $W_{\leq t} := \arg\min\{\widehat{\mathcal{R}}(W_i) : i \leq t, \|W_i - W_0\| \leq R_{\mathrm{gd}}\}$. Define effective radius $B := \min\left\{R_{\mathrm{gd}},\ 3R_Z + 2e\sqrt{\eta t\widehat{\mathcal{R}}^{(0)}(Z)}\right\}$, and linearization and generalization errors*

$$\tau := \frac{25\rho B^{4/3}\sqrt{d\ln(em^2 d^3/\delta)}}{m^{1/6}}, \qquad \tau_n := \frac{80\left(d\ln(em^2 d^3/\delta)\right)^{3/2}}{\sqrt{n}},$$

*and suppose $\tau \leq 2$. Then, with probability at least $1 - 3n\delta$, the selected iterate $W_{\leq t}$ satisfies $\|W_{\leq t} - W_0\| \leq B$, along with the empirical risk guarantee*

$$\widehat{\mathcal{R}}(W_{\leq t}) \leq e^{2\tau}\widehat{\mathcal{R}}^{(0)}(Z) + e^{\tau}(\rho R_Z)^2\epsilon_{\mathrm{gd}},$$

*and by discarding an additional $16\delta$ failure probability, then $\widehat{\mathcal{R}}^{(0)}(Z) \leq \mathcal{R}^{(0)}(Z) + \rho R_Z \tau_n$, and*

$$\mathcal{R}(W_{\leq t}) \leq e^{4\tau}\mathcal{R}^{(0)}(Z) + e^{3\tau}(\rho R_Z)^2\epsilon_{\mathrm{gd}} + e^{4\tau}(B + R_Z)\rho\tau_n.$$

This version of the statement, unlike Theorem 1.1, features an arbitrary reference *matrix $Z$*. This is powerful, though it can be awkward, since $W_0$ is a random variable.

## 2.4 Approximation analysis, consistency, and the proof of Theorem 1.1

Rather than trying to reason about good predictors which may happen to be close to random initialization, the approach here is instead to start from deterministic predictors over the population (e.g., $\overline{U}_\infty$), and to use their structure to construct approximants near the initial iterate, the random matrix $W_0$. Specifically, the approach here is fairly brute force: given initial weights $W_0$ with rows $(w_{0,j}^\intercal)_{j=1}^m$, the rows $(\overline{u}_j)_{j=1}^m$ of the finite width reference matrix $\overline{U} \in \mathbb{R}^{m \times d}$ intended to mimic $\overline{U}_\infty$ (which is after all a mapping $\overline{U}_\infty : \mathbb{R}^d \to \mathbb{R}^d$) are simply

$$\overline{u}_j := \frac{a_j \overline{U}_\infty(w_{0,j})}{\rho\sqrt{m}} + w_{0,j}. \tag{1}$$

By construction, $\|\overline{U} - W_0\| \leq R/\rho$, where $R := \sup_v \|\overline{U}_\infty(v)\|$. To argue that $\mathcal{R}^{(0)}(\overline{U})$ and $\mathcal{R}(\overline{U}_\infty)$ are close, the abstract control over the sphere in Lemma B.3 is again used. Plugging this $\overline{U}$ into Lemma 2.2 and introducing $\mathcal{K}_{\mathrm{bin}}(p_y, \phi_\infty)$ via Lemma B.1 gives the first part of Theorem 1.1, and the second part of Theorem 1.1 is also from Lemma B.1.

It remains to prove that for any $p_y$, there exists $\overline{U}_\infty$ with $\phi(x \mapsto f((x, 1)/\sqrt{2}; \overline{U}_\infty)) \approx p_y$ (we must include a bias term, as mentioned in Remark 1.1). If $p_y$ were continuous, there is a variant of Barron's seminal universal approximation construction which explicitly gives an infinite-width network of the desired form [Barron, 1993, Ji et al., 2020b]. To address continuity is even easier: *Lusin's theorem* [Folland, 1999, Theorem 7.10] lets us take the measurable function $p_y$, and obtain a continuous function that agrees with it on all but a negligible fraction of the domain. This completes the proof.

As mentioned, a key property of the reference model $\overline{U}_\infty$ is that it depends on neither the random sampling of data, nor the random sampling of weights. This vastly simplifies the proof of consistency, where the proof scheme first fixes an $\epsilon > 0$ and chooses a $\overline{U}_\infty$, and leaves it fixed as $m$ and $n$ vary.

**Corollary 2.3.** *Let early stopping parameter $\xi \in (0, 1)$ be given, and for each sample size $n$, define a weight matrix $\widehat{W}_n \in \mathbb{R}^{m^{(n)} \times (d+1)}$ and corresponding conditional probability model $\widehat{\phi}_n(x) := \phi(f((x, 1)/\sqrt{2}; \widehat{W}_n))$ as follows. For each sample size $n$, let $(W_i^{(n)})_{i \geq 0}$ denote the corresponding sequence of gradient descent iterates obtained with parameter choices $\rho^{(n)} := (m^{(n)})^{-1/8}$, and*

$m^{(n)} := n^{\frac{40}{3}(1-\xi)}$, and $\eta^{(n)} := 4/(\rho^{(n)})^2$, and $\epsilon_{\mathrm{gd}}^{(n)} := n^{\xi-1}$, and $t^{(n)} := n^{1-\xi}/8$, and choose the empirical risk minimizer over the sequence, meaning $\widehat{W}_n := \arg\min \{\widehat{\mathcal{R}}(W_i^{(n)}) : i \leq t^{(n)}\}$ (in the notation of Theorem 1.1, this is $W_{\leq t}$ with $R_{\mathrm{gd}} = \infty$). Then

$$\mathcal{R}(\widehat{W}_n) \longrightarrow \overline{\mathcal{R}} \text{ a.s.}, \qquad \mathcal{R}_{\mathrm{z}}(\widehat{W}_n) \longrightarrow \overline{\mathcal{R}}_{\mathrm{z}} \text{ a.s.}, \qquad \widehat{\phi}_n \xrightarrow{L_2(\mu_x)} p_y \text{ a.s.},$$

where the last convergence is in the $L_2(\mu_x)$ metric.

The use of a parameter $\xi \in (0,1)$ is standard in similar consistency results; see for instance the analogous parameter in the consistency analysis of AdaBoost [Bartlett and Traskin, 2007]. Proofs, as usual, are in the appendices.

## 3 Discussion and proof sketch of Proposition 1.2

Proposition 1.2 asserts that univariate *local interpolation rules* — predictors which perfectly fit the data, and are not too wild between data points of the same label — will necessarily achieve suboptimal population risk. The proof idea seems simple enough: if the true conditional probability $p_y$ is not one of $\{0, 1/2, 1\}$ everywhere, and is also continuous, then there must exist a region where it is well separated from these three choices. It seems natural that a constant fraction of the data in these regions will form adjacent pairs with the wrong label; a local interpolation rule will fail on exactly these adjacent noisy pairs, which suffices to give the bound. In reality, while this is indeed the proof scheme followed here, the full proof must contend with many technicalities and independence issues. It appears in the appendices.

While the motivation in Section 1.2 focused on neural networks which interpolate, and also maximum margin solutions, the behavior on this noisy univariate data is also well-illustrated by $k$-nearest-neighbors classifiers ($k$-nn). Specifically, 1-nn is a local interpolant, and Proposition 1.2 applies. On the other hand, choosing $k = \Theta(\ln(n))$ is known to provide enough smoothing to achieve consistency and avoid interpolation [Devroye et al., 1996].

It should be stressed again that even if the remaining pieces could be proved to apply this result to neural networks, namely necessitating early stopping, it would still be a univariate result only, leaving open many interesting possibilities in higher dimensions.

## 4 Concluding remarks and open problems

**Empirical performance.** Does the story here match experiments? E.g., is it often the case that if a neural network performs well, then so does a random feature model? Do neural networks fail on noisy data if care is not taken with temperature and early stopping? Most specifically, is this part of what happens in existing results reporting such failures [Guo et al., 2017]?

**Temperature parameter $\rho$.** Another interesting point of study is the temperature parameter $\rho$. It arises here in a fairly technical way: if $p_y$ is often close to $1/2$, then the random initialization of $W_0$ gets in the way of learning $p_y$. The temperature $\rho$ is in fact a brute-force method of suppressing this weight initialization noise. On the other hand, temperature parameters are common across many works which rely heavily on the detailed real-valued outputs of sigmoid and softmax mappings; e.g., in the distillation literature [Hinton et al., 2015]. The temperature also plays the same role as the scale parameter in the *lazy training* regime [Chizat and Bach, 2019]. Is $\rho$ generally useful, and does the analysis here relate to its practical utility?

**Random features, and going beyond the NTK.** The analysis here early stops before the feature learning begins to occur. How do things fare outside the NTK? Is there an analog of Theorem 1.1, still stopping shy of the interpolation pitfalls of Proposition 1.2, but managing to beat random features with some generality?

**The logistic loss.** One reason the logistic is used here is its simple interplay with calibration (e.g., see the elementary proof of Lemma B.1, as compared with the full machinery of classification calibration [Zhang, 2004, Bartlett et al., 2006]). The other key reason was the multiplicative error property Lemma B.1. Certainly, the logistic loss is widely used in practice; are the preceding technical points at all related to the widespread empirical use of the logistic loss?

## Acknowledgments and Disclosure of Funding

The authors are grateful for support from the NSF under grant IIS-1750051. MT thanks many friends for illuminating and motivating discussions: Daniel Hsu, Phil Long, Maxim Raginsky, Fanny Yang.

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
