# A Missing figures

Due to space limitations, Figures 1 and 2, referenced in the body, have been moved to this initial appendix section.

# B Proof of Theorem 1.1 and supporting results

This appendix section proves all bounds necessary for Theorem 1.1, and also proves the consistency statement in Corollary 2.3.

## B.1 Technical preliminaries

First, the key logistic loss properties.

**Lemma B.1.** *1. For any $a \geq b$,*

$$\frac{\phi(a)}{\phi(b)} \leq e^{a-b} \qquad \text{and} \qquad \frac{\ell(-a)}{\ell(-b)} \leq e^{a-b}.$$

*In particular, for any $f, g$ with $\sup_{\|x\| \leq 1} |f(x) - g(x)| \leq \tau$,*

$$e^{-\tau} \mathcal{R}(f) \leq \mathcal{R}(g) \leq e^{\tau} \mathcal{R}(f).$$

*If only $\max_k |f(x_k) - g(x_k)| \leq \tau$, then $e^{-\tau} \widehat{\mathcal{R}}(f) \leq \widehat{\mathcal{R}}(g) \leq e^{\tau} \widehat{\mathcal{R}}(f)$.*

*2. For any $f : \mathbb{R}^d \to \mathbb{R}$ and corresponding conditional model $\phi_f(x) := \phi(f(x))$,*

$$\frac{1}{2} \left( \mathcal{R}_{\mathsf{z}}(f) - \overline{\mathcal{R}}_{\mathsf{z}} \right)^2 \leq 2 \int (\phi_f(x) - p_y(x))^2 \, \mathrm{d}\mu_x(x) \leq \mathcal{K}_{\mathsf{bin}}(p_y, \phi_f) = \mathcal{R}(f) - \overline{\mathcal{R}}.$$

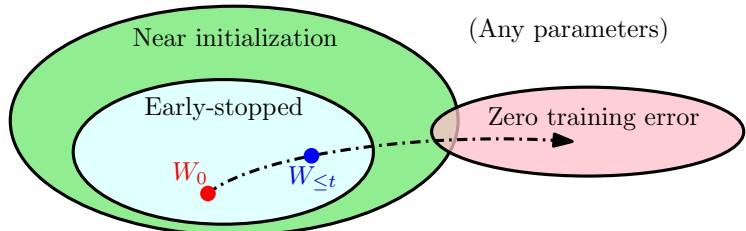

Figure 1: The setting of this paper, contrasted with standard settings. Theorem 1.1 considers iterate $W_{\leq t}$, which is somewhere in the *early-stopped* ball around the initial random choice $W_0$. This early-stopped ball is well inside the *near initialization* or *NTK* ball, since in noisy settings, the early-stopped ball will not reach zero training error, whereas the NTK ball will. Meanwhile, the NTK itself requires early stopping and is a subset of the space of all parameters.

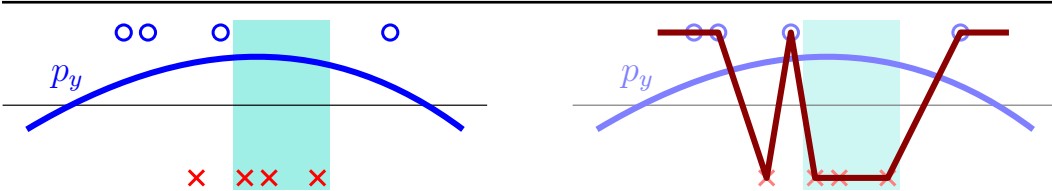

(a) Conditional model $p_y$ and some noisy data. A smoothed prediction rule would perform well.

(b) A *local interpolation rule* working very hard to fit the noisy data.

Figure 2: When data is noisy, it's best to give up on a few points. The shaded region here highlights consecutive points with the wrong label; as in Proposition 1.2, prediction rules that locally interpolate will have a large population risk in these regions.

*Proof.* 1. Since $a \geq b$, then $e^{b-a} \leq 1$, and

$$\frac{\phi(a)}{\phi(b)} = \frac{1 + e^{-b}}{1 + e^{-a}} = e^{a-b}\left(\frac{e^{b-a} + e^{-a}}{1 + e^{-a}}\right) \leq e^{a-b},$$

whereby

$$\int_{-\infty}^{a} \phi(r)\,\mathrm{d}r = \int_{-\infty}^{b} \phi(r + (a - b))\,\mathrm{d}r \leq e^{a-b}\int_{-\infty}^{b} \phi(r)\,\mathrm{d}r.$$

Consequently,

$$\ell(-a) = -\int_{-a}^{\infty} \ell'(r)\,\mathrm{d}r = \int_{-a}^{\infty} \phi(-r)\,\mathrm{d}r = \int_{-\infty}^{a} \phi(r)\,\mathrm{d}r \leq e^{a-b}\int_{-\infty}^{b} \phi(r)\,\mathrm{d}r = e^{a-b}\ell(-b).$$

The first set of claims for risk follow from the fact that for any pair $(x, y)$ and $\tau \geq 0$,

$$\ell(yr + y^2\tau) \leq \ell(yr) \leq \ell(yr - y^2\tau),$$

whereby

$$\mathcal{R}(f) = \mathbb{E}\ell(yf(x)) \leq \mathbb{E}\ell(yg(x) - \tau) \leq e^{\tau}\mathbb{E}\ell(yg(x)) = e^{\tau}\mathcal{R}(g).$$

The proof for empirical risk is similar, but only relies upon behavior on the finite sample.

2. From standard results in the literature on classification calibration [Zhang, 2004, Bartlett et al., 2006], the optimal logistic loss pointwise satisfies

$$\bar{r}_x := \inf_{r \in \mathbb{R}} p_y(x)\ell(r) + (1 - p_y(x))\ell(-r) = -p_y(x)\ln p_y(x) - (1 - p_y(x))\ln(1 - p_y(x)).$$

Consequently, for any predictor $f : \mathbb{R}^d \to \mathbb{R}$ and corresponding probability model $\phi_f(x) := \phi(f(x))$, note that

$$\mathcal{R}(f) = \int \big(p_y(x)\ln(1 + \exp(-f(x))) + (1 - p_y(x))\ln(1 + \exp(f(x)))\big)\,\mathrm{d}\mu_x(x)$$

$$= \int \big(-p_y(x)\ln\phi_f(x) - (1 - p_y(x))\ln(1 - \phi_f(x))\big)\,\mathrm{d}\mu_x(x),$$

and thus

$$\mathcal{R}(f) - \overline{\mathcal{R}} = \mathcal{K}_{\mathrm{bin}}(p_y, \phi_f).$$

By Pinsker's inequality,

$$\mathcal{K}_{\mathrm{bin}}(p_y, \phi_f) = \int \left(p_y(x)\ln\frac{p_y(x)}{\phi_f(x)} + (1 - p_y(x))\ln\frac{1 - p_y(x)}{1 - \phi_f(x)}\right)\,\mathrm{d}\mu_x(x)$$

$$\geq \frac{1}{2}\int \big(|p_y(x) - \phi_f(x)| + |(1 - p_y(x)) + (1 - \phi_f(x))|\big)^2\,\mathrm{d}\mu_x(x)$$

$$= 2\int \big(p_y(x) - \phi_f(x)\big)^2\,\mathrm{d}\mu_x(x).$$

If $\mathrm{sgn}(\phi_f(x) - 1/2) \neq \mathrm{sgn}(p_y(x) - 1/2)$, then $|\phi_f(x) - p_y(x)| \geq |p_y(x) - 1/2|$, and so

$$\mathcal{R}_{\mathrm{z}}(f) - \overline{\mathcal{R}}_{\mathrm{z}} = \int \mathbb{1}[\mathrm{sgn}(\phi_f(x) - 1/2) \neq \mathrm{sgn}(p_y(x) - 1/2)] \cdot |2p_y(x) - 1|\,\mathrm{d}\mu_x(x)$$

$$\leq 2\int |\phi_f(x) - p_y(x)|\,\mathrm{d}\mu_x(x)$$

$$\leq 2\sqrt{\int (\phi_f(x) - p_y(x))^2\,\mathrm{d}\mu_x(x)}.$$

$\square$

The remainder of this technical subsection develops a variety of concentration inequalities used throughout, most notably the control over the sphere in Lemma B.3. First, a few standard Gaussian inequalities, included here for completeness.

**Lemma B.2.** *Suppose $W \in \mathbb{R}^{m \times d}$ has iid Gaussian entries $W_{j,k} \sim \mathcal{N}(0,1)$, and let $(w_j^\mathsf{T})_{j=1}^m$ denote the rows.*

1. *For any $\tau > 0$, with probability at least $1 - 3\delta$,*
$$\sum_{j=1}^m \mathbb{1}\left[|w_j^\mathsf{T} x| \leq \tau \|x\|\right] \leq m\tau + \sqrt{8m\tau \ln(1/\delta)}.$$

2. *With probability at least $1 - \delta$,*
$$\|W\|_2 < \sqrt{m} + \sqrt{d} + \sqrt{2\ln(1/\delta)}.$$

3. *With probability at least $1 - 2\delta$,*
$$-\|z\|\sqrt{2\ln(1/\delta)} \leq \|\sigma_\mathrm{r}(Wz)\| - \mathbb{E}\|\sigma_\mathrm{r}(Wz)\| \leq \|z\|\sqrt{2\ln(1/\delta)},$$
*where*
$$\|z\|\left(\sqrt{\frac{m}{2}} - \frac{5}{\sqrt{8m}}\right) \leq \mathbb{E}\|\sigma_\mathrm{r}(Wz)\| \leq \|z\|\sqrt{\frac{m}{2}}.$$

4. *With probability at least $1 - \delta$, $w \in \mathbb{R}^d$ with coordinates $w_i \sim \mathcal{N}(0,1)$ satisfies*
$$\|w\| \leq \sqrt{d} + \sqrt{2\ln(1/\delta)}.$$

*Proof.*    1. For any row $j$, define an indicator random variable
$$P_j := \mathbb{1}[|w_j^\mathsf{T} x| \leq \tau \|x\|].$$
By rotational invariance, $P_j = \mathbb{1}[|w_{j,1}| \leq \tau]$, which by the form of the Gaussian density gives
$$\Pr[P_j = 1] \leq \frac{2\tau}{\sqrt{2\pi}} \leq \tau.$$
As such, by a multiplicative Chernoff bound [Blum et al., 2020, Theorem 12.6], with probability at least $1 - 3\delta$,
$$\sum_{j=1}^m P_j \leq m\Pr[P_1 = 1] + \sqrt{8m\Pr[P_1 = 1]\ln(1/\delta)} \leq m\tau + \sqrt{8m\tau\ln(1/\delta)},$$
as desired.

2. This is a standard spectral norm concentration bound for Gaussian matrices [Davidson and Szarek, 2001, Theorem II.13],

3. For the expectation, first note for a single row $w^\mathsf{T}$ by rotational invariance of the Gaussian that
$$\mathbb{E}\sigma_\mathrm{r}(w^\mathsf{T} x)^2 = \|x\|^2 \mathbb{E}\sigma_\mathrm{r}(w_1)^2 = \frac{1}{2}\|x\|^2 \mathbb{E}w_1^2 = \frac{\|x\|^2}{2}.$$
As such, for a full matrix $W$, the expected norm can be upper bounded via
$$\mathbb{E}\|\sigma_\mathrm{r}(Wx)\| \leq \sqrt{\mathbb{E}\|\sigma_\mathrm{r}(Wx)\|^2} = \sqrt{\frac{1}{2}\sum_{i=1}^m \|x\|^2} = \|x\|\sqrt{m/2},$$
and by a second-order lower bound, letting $\tilde{x} = x/\|x\|$ for convenience, and dividing through by $\sqrt{m/2}$ to ease notation,
$$\begin{aligned}
\mathbb{E}\sqrt{2\|\sigma_\mathrm{r}(Wx)\|^2/m} &= \|x\|\mathbb{E}\sqrt{2\|\sigma_\mathrm{r}(W\tilde{x})\|^2/m} \\
&\geq \|x\|\mathbb{E}\left(1 + (2\|\sigma_\mathrm{r}(W\tilde{x})\|^2/m - 1)/2 - (2\|\sigma_\mathrm{r}(W\tilde{x})\|^2/m - 1)^2/2\right) \\
&= \|x\|\left(1 - \mathbb{E}(2\|\sigma_\mathrm{r}(W\tilde{x})\|^2/m - 1)^2/2\right) \\
&= \|x\|\left(\frac{3}{2} - \frac{m(m-1)}{2m^2} - \frac{6m}{2m^2}\right) \\
&= \|x\|\left(1 - \frac{5}{2m}\right).
\end{aligned}$$

For the concentration part, note firstly that $\sigma_{\mathrm{r}}$ is $\ell_2$-Lipschitz when applied coordinate-wise, since

$$\|\sigma_{\mathrm{r}}(u) - \sigma_{\mathrm{r}}(v)\|^2 = \sum_{i=1}^m (\sigma_{\mathrm{r}}(u_i) - \sigma_{\mathrm{r}}(v_i))^2 \le \sum_{i=1}^m (u_i - v_i)^2 = \|u - v\|^2,$$

and thus

$$\|\sigma_{\mathrm{r}}(Ax)\| - \|\sigma_{\mathrm{r}}(Bx)\| \le \big\|\sigma_{\mathrm{r}}(Ax) - \sigma_{\mathrm{r}}(Bx)\big\| \le \|Ax - Bx\| \le \|A - B\|\,\|x\|,$$

and thus by standard Gaussian concentration, with probability at least $1 - \delta$,

$$\|\sigma_{\mathrm{r}}(Wx)\| - \mathbb{E}\|\sigma_{\mathrm{r}}(Wx)\| \le \|x\|\sqrt{2\ln(1/\delta)},$$

and vice versa.

4. This is a subset of the preceding proof: $w \mapsto \|w\|$ is 1-Lipschitz, thus by standard Gaussian concentration, with probability at least $1 - \delta$,

$$\|w\| \le \mathbb{E}\|w\| + \sqrt{2\ln(1/\delta)},$$

where $\mathbb{E}\|w\| \le \sqrt{\mathbb{E}\|w\|^2} = \sqrt{d}.$

$\square$

Next, finally, the control over the sphere, Lemma B.3. This lemma perhaps looks a bit underwhelming or simply abstract or overly complicated, but is a key tool in many steps of the proofs here; in particular, since it allows consideration for all $\|x\| \le 1$, it may be applied over the distribution. This consideration over the entire sphere contrasts this lemma (and its applications) from similar inequalities in prior work [Allen-Zhu et al., 2018b, Chen et al., 2021].

**Lemma B.3.** *Let scalars $R_V \ge 0$, and $\epsilon \in (0, 1/(md))$, and $m \ge \ln(edm)$ be given, along with a filter set $\mathcal{S}_0 \subseteq \mathbb{R}^{m \times d}$, and define $\mathcal{S} := \mathcal{S}_0 \cap \{V \in \mathbb{R}^{m \times d} : \|V - W_0\| \le R_V\}$. Let a function $h_V : \mathbb{R}^d \to \mathbb{R}$ be given with parameter $V \in \mathcal{S}$, and define functions*

$$\mathcal{H} := \left\{ x \mapsto h_V(x) + \left\langle \nabla f(x; W_0), V - W_0 \right\rangle : V \in \mathcal{S} \right\}.$$

*Moreover, let additional scalars $r_1, r_2, \delta$ satisfy the following conditions.*

1. *For every $x$ and $z$ with $\|x - z\| \le \epsilon$, then $\sup_{V \in \mathcal{S}} |h_V(x) - h_V(z)| \le r_1$.*

2. *For any fixed $\|x\| \le 1$, with probability at least $1 - \delta$, then $\sup_{h \in \mathcal{H}} |h(x)| \le r_2$.*

*Then with probability at least $1 - (\sqrt{d}/\epsilon)^d \delta$,*

$$\sup_{\|x\| \le 1} \sup_{h \in \mathcal{H}} |h(x)| \le r_2 + r_1 + 11 R_V \rho \left( \frac{\ln(edm/\delta)}{m} \right)^{1/4}.$$

The proof of Lemma B.3 will need two technical lemmas. The first is a basic property of inner products and arccosine which also makes a later appearance in Lemma B.11.

**Lemma B.4.** *If $\|x - z\| \le \epsilon$ and $x, z \ne 0$, then*

$$1 \ge \left\langle \frac{x}{\|x\|}, \frac{z}{\|z\|} \right\rangle \ge 1 - \frac{2\epsilon^2}{\|x\|^2}, \qquad \text{and} \qquad \arccos\left( \left\langle \frac{x}{\|x\|}, \frac{z}{\|z\|} \right\rangle \right) \le \frac{\epsilon\sqrt{8}}{\|x\|}.$$

*Proof.* The first inequalities follow from

$$1 \geq \left\langle \frac{x}{\|x\|}, \frac{z}{\|z\|} \right\rangle$$

$$= 1 - \frac{1}{2} \left\| \frac{x}{\|x\|} - \frac{z}{\|z\|} \right\|^2$$

$$= 1 - \frac{1}{2\|x\|^2\|z\|^2} \Big\| \|x\|z\| - z\|z\| + z(\|z\| - \|x\|) \Big\|^2$$

$$\geq 1 - \frac{\|x-z\|^2\|z\|^2 + \|z\|^2(\|z\| - \|x\|)^2}{\|x\|^2\|z\|^2}$$

$$\geq 1 - \frac{2\|x-z\|^2\|z\|^2}{\|x\|^2\|z\|^2}$$

$$\geq 1 - \frac{2\epsilon^2}{\|x\|^2}.$$

To finish, since $\arccos$ is decreasing along $[0, 1]$, and since for any $a \in [0, 1]$,

$$\arccos(1 - a) = \int_{1-a}^{1} \frac{\mathrm{d}r}{\sqrt{1 - r^2}} = \int_0^a \frac{\mathrm{d}r}{\sqrt{2r - r^2}} \leq \int_0^a \frac{\mathrm{d}r}{\sqrt{r}} = 2\sqrt{a},$$

then

$$\arccos\left( \left\langle \frac{x}{\|x\|}, \frac{z}{\|z\|} \right\rangle \right) \leq \arccos\left( 1 - \frac{2\epsilon^2}{\|x\|^2} \right) \leq 2\sqrt{\frac{2\epsilon^2}{\|x\|^2}} = \frac{\epsilon\sqrt{8}}{\|x\|}.$$

$\square$

The main heavy lifting in Lemma B.3 is encapsulated in the following concentration inequality. In words, it controls the behavior of the initial features within a tiny localized region of the sphere; the proof of Lemma B.3 combines this local control with a discrete cover of the sphere, together giving control over the entire sphere.

**Lemma B.5.** *Let any fixed $\|z\| \leq 1$ be given (independent of $W_0$), along with a scalar $\epsilon > 0$ with $\epsilon \leq 1/(dm)$, where $m \geq \ln(edm)$. Then, with probability at least $1 - \delta$,*

$$\sup_{\substack{\|x-z\| \leq \epsilon \\ \|x\| \leq 1}} \|\nabla f(x; W_0) - \nabla f(z; W_0)\|^2 \leq 113\rho^2 \sqrt{\frac{\ln(edm/\delta)}{m}}.$$

*Proof.* Throughout the proof, simplify notation via $W := W_0$, and let $(w_j^\mathsf{T})_{j=1}^m$ denote the rows of $W$, and furthermore write

$$g(x, z; w) := \frac{\rho^2}{m} \Big\| x\mathbb{1}[w_j^\mathsf{T}x \geq 0] - z\mathbb{1}[w_j^\mathsf{T}z \geq 0] \Big\|^2.$$

Lastly, for any $x \in \mathbb{R}^d$ under consideration, then $\|x\| \leq 1$, so this condition will often be implicit. Note that

$$\sup_{\|x-z\| \leq \epsilon} \|\nabla f(x; W) - \nabla f(z; W)\|^2 = \frac{\rho^2}{m} \sup_{\|x-z\| \leq \epsilon} \sum_{j=1}^m \|x\mathbb{1}[w_j^\mathsf{T}x \geq 0] - z\mathbb{1}[w_j^\mathsf{T}z \geq 0]\|^2$$

$$= \sup_{\|x-z\| \leq \epsilon} \sum_{j=1}^m g(x, z; w_j).$$

Next note that this quantity, treated as a function of the $m$ rows of $W$, satisfies bounded differences with constant $\rho^2/m$: letting $W'$ be a copy of $W$ which differs only in a single row $w_i'$, and noting

$g \geq 0$,

$$\left| \sup_{\|x-z\|\leq\epsilon} \sum_{j=1}^m g(x,z;w_j) - \sup_{\|x-z\|\leq\epsilon} \sum_{j=1}^m g(x,z;w_j') \right|$$

$$= \left| \sup_{\|x-z\|\leq\epsilon} \sum_{j=1}^m g(x,z;w_j) - \sup_{\|x-z\|\leq\epsilon} \left( g(x,z;w_i) - g(x,z;w_i) + \sum_{j=1}^m g(x,z;w_j') \right) \right|$$

$$\leq \sup_{\|x-z\|\leq\epsilon} \left| g(x,z;w_i') - g(x,z;w_i) \right| \leq \frac{\rho^2}{m}.$$

As such, by McDiarmid's inequality, with probability at least $1-\delta$,

$$\sup_{\|x-z\|\leq\epsilon} \|\nabla f(x;W) - \nabla f(z;W)\|^2 \leq \sqrt{\rho^4 \ln(1/\delta)/(2m)}$$

$$+ \mathbb{E}_W \sup_{\|x-z\|\leq\epsilon} \|\nabla f(x;W) - \nabla f(z;W)\|^2. \quad (2)$$

It remains to analyze this expectation. First consider the case that $\|z\| \leq 3\sqrt{\epsilon}$; then, for any $W$,

$$\sup_{\|x-z\|\leq\epsilon} \|\nabla f(x;W) - \nabla f(z;W)\|^2 = \frac{\rho^2}{m} \sup_{\|x-z\|\leq\epsilon} \sum_{j=1}^m \|x\mathbb{1}[w_j^\mathsf{T}x \geq 0] - z\mathbb{1}[w_j^\mathsf{T}z \geq 0]\|^2$$

$$\leq \frac{2\rho^2}{m} \sup_{\|x-z\|\leq\epsilon} \sum_{j=1}^m \left( \|x\|^2 + \|z\|^2 \right)$$

$$\leq \frac{2\rho^2}{m} \sum_{j=1}^m (16\epsilon + 9\epsilon) \leq 50\epsilon\rho^2. \quad (3)$$

For the rest of the proof, suppose $\|z\| > 3\sqrt{\epsilon}$, which also implies $\|x\| > 2\sqrt{\epsilon}$ for every $x$ satisfying $\|x - z\| \leq \epsilon$.

Since $z$ is fixed, and in particular does not depend on $W$, we may use the rotational invariance of $W$ to leverage the condition $\|x - z\| \leq \epsilon$. Specifically, define a matrix $M \in \mathbb{R}^{d\times d}$ whose first column is $z/\|z\|$, and the remaining columns are orthonormal (we can not use $x$ in the definition of $M$, since $x$ varies within the expectation). Defining (for any $x$) the two projections $x_z := zx^\mathsf{T}z/\|z\|^2$ and $x^\perp := x - x_z$ (whereby $z^\mathsf{T}x^\perp = 0$), we may rotate the rows of $W$ by $M$, giving

$$\mathbb{1}\left[(Mw_j)^\mathsf{T}z \geq 0\right] = \mathbb{1}\left[w_{j,1}\|z\| \geq 0\right]$$

$$= \mathbb{1}\left[w_{j,1} \geq 0\right],$$

$$\mathbb{1}\left[(Mw_j)^\mathsf{T}x \geq 0\right] = \mathbb{1}\left[w_j^\mathsf{T}M^\mathsf{T}(x_z + x^\perp) \geq 0\right]$$

$$= \mathbb{1}\left[w_{j,1}z^\mathsf{T}x/\|z\| \geq -w_j^\mathsf{T}M^\mathsf{T}x^\perp\right]$$

$$= \mathbb{1}\left[w_{j,1} \geq -\frac{w_j^\mathsf{T}M^\mathsf{T}x^\perp}{z^\mathsf{T}x/\|z\|}\right],$$

where the last division does not change the sign due to $\|z - x\| \leq \epsilon$ and $\|z\| > 3\sqrt{\epsilon}$, for instance as verified by upcoming invocations of Lemma B.4. Now let $E_j$ denote the event that for this $w_j$, there exists $\|x - z\| \leq \epsilon$ such that these two indicators are not equal. Letting $\tau > 0$ denote a free parameter to be optimized later, this event is implied by the union of two simpler events: let $w_{j,2:} \in \mathbb{R}^{d-1}$ denote all but the first coordinate of $w_j$, and define

$$E_{j,1} := \left[|w_{j,1}| \leq \tau\right], \qquad E_{j,2} := \left[\sup_{\|x-z\|\leq\epsilon} \frac{\|w_{j,2:}\| \cdot \|x^\perp\| \cdot \|z\|}{z^\mathsf{T}x} > \tau\right];$$

by construction (and Cauchy-Schwarz), if the negation of both events holds, then the indicators are the same. To upper bound the probability of the first event, by the form of the Gaussian density,

$$\Pr[E_{j,1}] \leq \tau\sqrt{\frac{2}{\pi}} < \tau.$$

To control the various terms in $E_{j,2}$, firstly by Lemma B.2, with probability at least $1 - \epsilon$, then

$$\|w_{j,2:}\| \leq \sqrt{d-1} + \sqrt{2\ln(1/\epsilon)} \leq \sqrt{2d - 2 + 4\ln(1/\epsilon)};$$

this will be the only step of the derivation controlling $\Pr[E_{j,2}]$, and note that it depends only on $w_j$ and $z$ and not on any specific $x$. Next, by Lemma B.4, for any $\|x - z\| \leq \epsilon$, since $\|x\| \geq 2\epsilon$ (whereby $2\epsilon^2/\|x\|^2 < 1$),

$$\|x^\perp\|^2 = \|x\|^2 - \frac{(z^\intercal x)^2}{\|z\|^2} = \|x\|^2 \left(1 - \left[\frac{z^\intercal x}{\|x\|\|z\|}\right]^2\right)$$

$$\leq \|x\|^2 \left(1 - \left[1 - \frac{2\epsilon^2}{\|x\|^2}\right]^2\right) = 4\epsilon^2 - \frac{4\epsilon^4}{\|x\|^2} \leq 4\epsilon^2.$$

Similarly by Lemma B.4, using $\epsilon \leq 1$,

$$\frac{z^\intercal x}{\|z\|} \geq \|x\| - \frac{2\epsilon^2}{\|x\|} > 2\sqrt{\epsilon} - \epsilon^{1.5} \geq \sqrt{\epsilon}.$$

Combining all these pieces, with probability at least $1 - \epsilon$,

$$\frac{\|w_{j,2:}\| \cdot \|x^\perp\| \cdot \|z\|}{z^\intercal x} \leq \sqrt{2d - 2 + 4\ln(1/\epsilon)} \left(\frac{2\epsilon}{\sqrt{\epsilon}}\right) \leq 4\sqrt{d\epsilon \ln(e/\epsilon)}.$$

This right hand side does not depend on the specific choice of $x$, and holds for any $\|x - z\| \leq \epsilon$. As such, set $\tau := 4\sqrt{d\epsilon \ln(e/\epsilon)}$, whereby

$$\Pr[E_j] \leq \Pr[E_{j,1}] + \Pr[E_{j,2}] \leq \tau + \epsilon.$$

Moreover, by a multiplicative Chernoff bound [Blum et al., 2020, Theorem 12.6], with probability at least $1 - 3\epsilon$, the events $(E_j)_{j=1}^m$ hold for at most $m_\tau := m(\tau + \epsilon) + \sqrt{8m(\tau + \epsilon)\ln(1/\epsilon)}$ rows. Now let $E_\tau$ denote the event that $(E_j)_{j=1}^m$ holds for at most $m_\tau$ rows. Then

$$\mathbb{E}_W \sup_{\|x-z\| \leq \epsilon} \|\nabla f(x; W) - \nabla f(z; W)\|^2.$$

$$= \mathbb{E}_W \left[\sup_{\|x-z\| \leq \epsilon} \|\nabla f(x; W) - \nabla f(z; W)\|^2 \mid E_\tau\right] \Pr[E_\tau]$$

$$+ \mathbb{E}_W \left[\sup_{\|x-z\| \leq \epsilon} \|\nabla f(x; W) - \nabla f(z; W)\|^2 \mid E_\tau^c\right] \Pr[E_\tau^c]$$

$$\leq 2\rho^2 \sup_{\|x-z\| \leq \epsilon} \left(\frac{m}{m}\|x - z\|^2 + \frac{m_\tau}{m}(\|x\|^2 + \|z\|^2)\right) + \sup_{\|x-z\| \leq \epsilon} \left(3\epsilon(\|x\|^2 + \|z\|^2)\right)$$

$$\leq 2\rho^2 \left(\epsilon^2 + \frac{2m_\tau}{m} + 6\epsilon\right). \tag{4}$$

The proof will now be completed by returning to the McDiarmid application resulting in eq. (2), and combining all preceding bounds. Starting with a simplification via the assumption $\epsilon \leq 1/(dm)$ and $m \geq \ln(edm)$, note

$$\tau = 4\sqrt{d\epsilon \ln(e/\epsilon)} \leq 4\sqrt{\frac{\ln(edm)}{m}},$$

$$\frac{m_\tau}{m} = \tau + \epsilon + \sqrt{8(\tau + \epsilon)\ln(1/\epsilon)/m}$$

$$\leq 5\sqrt{\frac{\ln(edm)}{m}} + \sqrt{\frac{40\sqrt{\ln(edm)}\ln(edm)}{m^{3/2}}} \leq 12\sqrt{\frac{\ln(edm)}{m}}.$$

Combining the preceding simplifications with eqs. (3) and (4), continuing from the McDiarmid application in eq. (2), with probability at least $1 - \delta$,

$$\sup_{\substack{\|x-z\| \leq \epsilon \\ \|x\| \leq 1}} \|\nabla f(x; W_0) - \nabla f(z; W_0)\|^2 \leq \rho^2 \left( \sqrt{\frac{\ln(1/\delta)}{2m}} + 50\epsilon + 2 \left( \epsilon^2 + \frac{2m_\tau}{m} + 6\epsilon \right) \right)$$

$$\leq \rho^2 \left( \sqrt{\frac{\ln(1/\delta)}{2m}} + \frac{50}{md} + \frac{2}{m^2 d^2} + 48\sqrt{\frac{\ln(edm)}{m}} + \frac{12}{md} \right)$$

$$\leq 113\rho^2 \sqrt{\frac{\ln(edm/\delta)}{m}}.$$

□

Finally, the proof of Lemma B.3 via the preceding technical lemmas.

*Proof of Lemma B.3.* Let $\mathcal{C}$ denote a cover of each coordinate of $\|x\| \leq 1$ at scale $\epsilon/\sqrt{d}$, meaning $|\mathcal{C}| \leq (\sqrt{d}/\epsilon)^d$ (the grid elements can be $2\epsilon/\sqrt{d}$ apart), and for any $\|x\| \leq 1$, there exists $z \in \mathcal{C}$ with

$$\|z - x\| = \sqrt{\sum_{i=1}^{d} (z_i - x_i)^2} \leq \epsilon.$$

This cover $\mathcal{C}$ will be used throughout the proof; it is crucial that its construction makes no reference to $W_0$, and in particular that the cover elements are independent of $W_0$.

Union bound together and discard $|\mathcal{C}|\delta$ failure probability so that for every $z \in \mathcal{C}$, then $\sup_{h \in \mathcal{H}} |h(z)| \leq r_2$. Additionally union bound together and discard $|\mathcal{C}|\delta$ failure probability corresponding to instantiating Lemma B.5 for each $z \in \mathcal{C}$, whereby

$$\max_{z \in \mathcal{C}} \sup_{\substack{\|x-z\| \leq \epsilon \\ \|x\| \leq 1}} \|\nabla f(x; W_0) - \nabla f(z; W_0)\|^2 \leq 113\rho^2 \sqrt{\frac{\ln(edm/\delta)}{m}}.$$

Now let an arbitrary $\|x\| \leq 1$ be given, and let $z \in \mathcal{C}$ be a nearest cover element, whereby $\|z - x\| \leq \epsilon$. Then

$$\sup_{h \in \mathcal{H}} |h(x)| \leq \sup_{h \in \mathcal{H}} \left( |h(z)| + |h(z) - h(x)| \right)$$

$$\leq r_2 + \sup_{V \in \mathcal{S}} |h_V(z) - h_V(x)| + \sup_{V \in \mathcal{S}} |\langle \nabla f(x; W_0) - \nabla f(z; W_0), V - W_0 \rangle|$$

$$\leq r_2 + r_1 + \sup_{V \in \mathcal{S}} \|\nabla f(x; W_0) - \nabla f(z; W_0)\| \cdot \|V - W_0\|$$

$$\leq r_2 + r_1 + 11 R_V \rho \left( \frac{\ln(edm/\delta)}{m} \right)^{1/4}.$$

□

As a first application of Lemma B.3, the range of the mappings can be bounded for all $\|x\| \leq 1$, which is used later in the generalization analysis.

**Lemma B.6.** *Let $R_V > 0$ be given.*

1. *For any $x \in \mathbb{R}^d$, with probability at least $1 - 3\delta$, every $V \in \mathbb{R}^{m \times d}$ satisfies*

$$\left| \langle \nabla f(x; W_0), V \rangle \right| \leq \rho \|x\| \left( \|V - W_0\|_{\mathrm{F}} + 2\ln(1/\delta) \right).$$

2. *Suppose $R_V \geq 1$ and $m \geq \ln(emd)$. With probability at least $1 - (1 + 3(md^{3/2})^d)\delta$,*

$$\sup_{\|V - W_0\| \leq R_V} \sup_{\|x\| \leq 1} \left| \langle \nabla f(x; W_0), V \rangle \right| \leq 18 R_V \rho \ln(emd/\delta).$$

*Proof.* For convenience throughout the proof, write $W := W_0$.

1. Splitting terms via $V = V - W + W$,

$$\left|\langle \nabla f(x;W), V\rangle\right| \leq \left|\langle \nabla f(x;W), W\rangle\right| + \left|\langle \nabla f(x;W), V - W\rangle\right|.$$

For the first term, since $W$ is independent of $a$ and can be treated as fixed, by Hoeffding's inequality, with probability at least $1 - 2\delta$ over the draw of $a$,

$$\left|\langle \nabla f(x;W), W\rangle\right| = \left|f(x;W)\right| \leq \frac{\rho}{\sqrt{m}}\|\sigma_{\mathrm{r}}(Wx)\|\sqrt{\ln(1/\delta)/2}.$$

By Lemma B.2, with additional failure probability $\delta$,

$$\|\sigma_{\mathrm{r}}(Wx)\| \leq \mathbb{E}\|\sigma_{\mathrm{r}}(Wx)\| + \|x\|\sqrt{2\ln(1/\delta)} \leq \|x\|\left(\sqrt{m/2} + \sqrt{2\ln(1/\delta)}\right).$$

Together,

$$\left|\langle \nabla f(x;W), W\rangle\right| \leq \rho\|x\|\left(1 + \sqrt{2\ln(1/\delta)/m}\right)\sqrt{\ln(1/\delta)/2}.$$

For the second term, due to the scale of the first term, it suffices to worst-case everything: by Cauchy-Schwarz,

$$\left|\langle \nabla f(x;W), V - W\rangle\right| \leq \|\nabla f(x;W)\|_{\mathrm{F}} \cdot \|V - W\|_{\mathrm{F}} \leq \rho\|x\| \cdot \|V - W\|_{\mathrm{F}}.$$

Combining everything, with probability at least $1 - 3\delta$,

$$\begin{aligned}\left|\langle \nabla f(x;W), V\rangle\right| &\leq \rho\|x\|\left(\|V - W\|_{\mathrm{F}} + \sqrt{\ln(1/\delta)/2} + \ln(1/\delta)/\sqrt{m}\right) \\ &\leq \rho\|x\|\left(\|V - W\|_{\mathrm{F}} + 2\ln(1/\delta)\right)\end{aligned}$$

2. This item proceeds by combining the previous item with the covering argument from Lemma B.3. Concretely, define the function

$$h_V(x) := f^{(0)}(x);$$

that is, $h_V$ has no dependence on $V \in \mathbb{R}^{m \times d}$, but note that

$$\langle \nabla f(x;W), V\rangle = \langle \nabla f(x;W), V - W\rangle + \langle \nabla f(x;W), W\rangle = \langle \nabla f(x;W), V - W\rangle + h_V(x),$$

which is precisely the expression controlled by Lemma B.3. Let $\mathcal{H}$ denote the class of functions defined there.

By the preceding item, for any fixed $\|x\| \leq 1$, with probability at least $1 - \delta$,

$$\sup_{h \in \mathcal{H}} |h(x)| \leq \rho\left(R_V + 2\ln(1/\delta)\right) =: r_2.$$

Moreover, by Lemma B.2, with probability at least $1 - \delta$, then $\|W\|_2 \leq \sqrt{m} + \sqrt{d} + \sqrt{2\ln(1/\delta)}$, thus for any $\|x - z\| \leq \epsilon$, with $\epsilon$ to be determined later,

$$\begin{aligned}|f(x;W) - f(z;W)| &\leq \frac{\rho}{\sqrt{m}}\|a\| \cdot \|W(x - z)\| \leq \rho\|W\|_2\|x - z\| \\ &\leq \rho(\sqrt{m} + \sqrt{d} + \sqrt{2\ln(1/\delta)})\epsilon =: r_1.\end{aligned}$$

As such, by Lemma B.3, choosing $\epsilon := 1/(md)$ and $\mathcal{S}_0 = \mathbb{R}^{m \times d}$, with probability at least $1 - 3(md^{3/2})^d\delta$,

$$\begin{aligned}\sup_{\|V - W\| \leq R_V} \sup_{\|x\| \leq 1} h_V(x) &\leq r_2 + r_1 + 11R_V\rho\left(\frac{\ln(edm/\delta)}{m}\right)^{1/4}. \\ &\leq \rho\left(R_V + 2\ln(1/\delta)\right) \\ &\quad + \rho(\sqrt{m} + \sqrt{d} + \sqrt{2\ln(1/\delta)})\epsilon \\ &\quad + 11R_V\rho\left(\frac{\ln(edm/\delta)}{m}\right)^{1/4}. \\ &\leq 18R_V\rho\ln(emd/\delta).\end{aligned}$$

$\square$

Next, the linear approximation bounds; the last two items use Lemma B.3 to control all points on the sphere. As mentioned before, this is in contrast to prior presentations of linear approximation inequalities, which only establish the bounds on the finite training sample [Chen et al., 2021, Allen-Zhu et al., 2018b]. Note that the bounds over the sphere have a more restrictive statement; the present proof does not handle the more general form presented for a finite sample.

**Lemma B.7** (See also Lemma 2.1). *Let scalars $\delta > 0$ and $R_V \geq 1$ and $R_B \geq 0$ be given.*

1. *For any fixed $x \in \mathbb{R}^d$, with probability at least $1 - 3\delta$, for any $V \in \mathbb{R}^{m \times d}$ and $B \in \mathbb{R}^{m \times d}$ with $\|V - W_0\| \leq R_V$ and $\|B - W_0\| \leq R_B$,*

$$\left| \langle \nabla f(x; V) - \nabla f(x; W_0), B \rangle \right| \leq \frac{3\rho \|x\| \left( R_B + 2R_V \right) R_V^{1/3} \ln(e/\delta)^{1/4}}{m^{1/6}} =: \tau_1.$$

2. *Let $\tau_1$ be as in the previous part. With probability at least $1 - 3n\delta$,*

$$\sup_{\|W_i - W_0\| \leq R_V} \sup_{\|W_j - W_0\| \leq R_V} \sup_{\|B - W_0\| \leq R_B} \frac{\widehat{\mathcal{R}}^{(i)}(B)}{\widehat{\mathcal{R}}^{(j)}(B)} \leq e^{2\tau_1}.$$

3. *Suppose $m \geq \ln(edm)$. With probability at least $1 - (1 + 3(d^2 m)^d)\delta$,*

$$\sup_{\|V - W_0\| \leq R_V} \sup_{\|x\| \leq 1} \left| \langle \nabla f(x; V) - \nabla f(x; W_0), V \rangle \right| \leq \frac{25\rho R_V^{4/3} \sqrt{\ln(edm/\delta)}}{m^{1/6}} =: \tau_3.$$

4. *Let $\tau_3$ be as in the previous part and again suppose $m \geq \ln(edm)$. With probability at least $1 - (1 + 3(d^2 m)^d)\delta$,*

$$\sup_{\|W_i - W_0\| \leq R_V} \frac{\mathcal{R}(W_i)}{\mathcal{R}^{(0)}(W_i)} \leq e^{\tau_3}.$$

*Proof of Lemmas 2.1 and B.7.* The first item implies the second via Lemma B.1, and moreover implies the third item via Lemma B.3. Similarly, the third item implies the fourth via Lemma B.1. Throughout the proof, write $W := W_0$ with rows $(w_j^\mathsf{T})_{j=1}^m$ for convenience.

1. Fix $x \in \mathbb{R}^d$. Fix a parameter $r > 0$, which will be optimized at the end of the proof. Let $V$ and $B$ be given with $\|V - W\| \leq R_V$ and $\|B - W\| \leq R_B$.

   Define the sets

$$\begin{aligned} S_1 &:= \left\{ j \in [m] : |w_j^\mathsf{T} x| \leq r \|x\| \right\}, \\ S_2 &:= \left\{ j \in [m] : \|v_j - w_j\| \geq r \right\} \\ S &:= S_1 \cup S_2. \end{aligned}$$

   By Lemma B.2, with probability at least $1 - 3\delta$,

$$|S_1| \leq rm + \sqrt{8rm \ln(1/\delta)}.$$

   On the other hand,

$$R_V^2 \geq \|V - W\|^2 \geq \sum_{j \in S_2} \|v_j - w_j\|^2 \geq |S_2| r^2,$$

   meaning $|S_2| \leq R_V^2 / r^2$. For any $j \notin S$, if $w_j^\mathsf{T} x > 0$, then

$$v_j^\mathsf{T} x \geq w_j^\mathsf{T} x - \|v_j - w_j\| \cdot \|x\| > \|x\| (r - r) = 0,$$

   meaning $\mathbb{1}[w_j^\mathsf{T} x \geq 0] = \mathbb{1}[v_j^\mathsf{T} x \geq 0]$; the case that $j \notin S$ and $w_j^\mathsf{T} x < 0$ is analogous. Together,

$$|S| \leq rm + \sqrt{8rm \ln(1/\delta)} + \frac{R_V^2}{r^2} \quad \text{and} \quad j \notin S \implies \mathbb{1}[w_j^\mathsf{T} x \geq 0] = \mathbb{1}[v_j^\mathsf{T} x \geq 0].$$

Continuing,

$$\frac{\sqrt{m}}{\rho}\left|\langle\nabla f(x;V)-\nabla f(x;W),B\rangle\right|$$

$$\leq \frac{\sqrt{m}}{\rho}\left|\langle\nabla f(x;V)-\nabla f(x;W),V\rangle\right| + \frac{\sqrt{m}}{\rho}\left|\langle\nabla f(x;V)-\nabla f(x;W),V-B\rangle\right|$$

$$= \left|a^{\mathsf{T}}\left(\operatorname{diag}(\mathbb{1}[V^{\mathsf{T}}x\geq 0])-\operatorname{diag}(\mathbb{1}[W^{\mathsf{T}}x\geq 0])\right)Vx\right|$$

$$+ \left|a^{\mathsf{T}}\left(\operatorname{diag}(\mathbb{1}[V^{\mathsf{T}}x\geq 0])-\operatorname{diag}(\mathbb{1}[W^{\mathsf{T}}x\geq 0])\right)(V-B)x\right|.$$

Handling these two terms separately, the second term is easier: by Cauchy-Schwarz,

$$\left|a^{\mathsf{T}}\left(\operatorname{diag}(\mathbb{1}[V^{\mathsf{T}}x\geq 0])-\operatorname{diag}(\mathbb{1}[W^{\mathsf{T}}x\geq 0])\right)(V-B)x\right| \leq \sqrt{|S|}\|(V-W-(B-W))x\|$$

$$\leq \sqrt{|S|}\,(R_V+R_B)\,\|x\|.$$

For the first term,

$$\left|a^{\mathsf{T}}\left(\operatorname{diag}(\mathbb{1}[V^{\mathsf{T}}x\geq 0])-\operatorname{diag}(\mathbb{1}[W^{\mathsf{T}}x\geq 0])\right)Vx\right| \leq \sum_{j=1}^{m}\mathbb{1}[\operatorname{sgn}(v_j^{\mathsf{T}}x)\neq \operatorname{sgn}(w_j^{\mathsf{T}}x)]\cdot|v_j^{\mathsf{T}}x|.$$

If $v_j^{\mathsf{T}}x$ and $w_j^{\mathsf{T}}x$ have different signs, then $|v_j^{\mathsf{T}}x|\leq |v_j^{\mathsf{T}}x-w_j^{\mathsf{T}}x|\leq \|v_j-w_j\|\cdot\|x\|$; plugging this in, by Cauchy-Schwarz,

$$\sum_{j=1}^{m}\mathbb{1}[\operatorname{sgn}(v_j^{\mathsf{T}}x)\neq \operatorname{sgn}(w_j^{\mathsf{T}}x)]\cdot|v_j^{\mathsf{T}}x| \leq \sum_{j=1}^{m}\mathbb{1}[\operatorname{sgn}(v_j^{\mathsf{T}}x)\neq \operatorname{sgn}(w_j^{\mathsf{T}}x)]\cdot\|v_j-w_j\|\cdot\|x\|$$

$$\leq \sum_{j\in S}\|v_j-w_j\|\cdot\|x\|$$

$$\leq \sqrt{|S|}\|V-W\|_{\mathrm{F}}\|x\|$$

$$\leq R_V\sqrt{|S|}\|x\|.$$

Combining these derivations,

$$\left|\langle\nabla f(x;V)-\nabla f(x;W),B\rangle\right| \leq \frac{\rho}{\sqrt{m}}\left(\sqrt{|S|}\,(R_V+R_B)\,\|x\|+R_V\sqrt{|S|}\|x\|\right)$$

$$\leq \frac{\rho\sqrt{|S|}\|x\|\,(2R_V+R_B)}{\sqrt{m}}.$$

Rearranging, and expanding the definition of $|S|$ with the choice $r:=R_V^{2/3}m^{-1/3}$, and using $R_V\geq 1$,

$$\left|\langle\nabla f(x;V)-\nabla f(x;W),B\rangle\right| \leq \frac{\rho\|x\|\,(R_B+2R_V)}{\sqrt{m}}\sqrt{rm+\sqrt{8rm\ln(1/\delta)}+\frac{R_V^2}{r^2}}$$

$$\leq \frac{\rho\|x\|\,(R_B+2R_V)\,R_V^{1/3}m^{1/3}\ln(e/\delta)^{1/4}}{\sqrt{m}}\sqrt{1+\sqrt{8}+1}$$

$$\leq \frac{3\rho\|x\|\,(R_B+2R_V)\,R_V^{1/3}\ln(e/\delta)^{1/4}}{m^{1/6}}.$$

2. Union bounding the previous part over all $(x_k)_{k=1}^{n}$, with probability at least $1-\delta$, for any iterations $(i,j)$ and for any matrices $(W_i,W_j,B)$ satisfying $\max\{\|W_i-W_0\|,\|W_j-W_0\|,\|B-W_0\|\}\leq R_V$

$$\max_{k}\left|\langle\nabla f(x_k;W_i)-\nabla f(x_k;W),B\rangle\right|\leq \tau_1.$$

In particular, by Lemma B.1,

$$e^{-\tau_1}\leq \frac{\widehat{\mathcal{R}}^{(i)}(B)}{\widehat{\mathcal{R}}^{(0)}(B)}\leq e^{\tau_1}.$$

Applying this twice gives

$$e^{-2\tau_1} \le \frac{\widehat{\mathcal{R}}^{(i)}(B)}{\widehat{\mathcal{R}}^{(0)}(B)} \left( \frac{\widehat{\mathcal{R}}^{(0)}(B)}{\widehat{\mathcal{R}}^{(j)}(B)} \right) = \frac{\widehat{\mathcal{R}}^{(i)}(B)}{\widehat{\mathcal{R}}^{(j)}(B)} \le e^{2\tau_1}.$$

3. This part follows from the first via Lemma B.3. As such, for every $\|V - W\| \le R_V$, define

$$h_V(x) := f(x; W) - f(x; V);$$

by this choice,

$$
\begin{aligned}
\langle \nabla f(x; V) - \nabla f(x; W), V \rangle &= \langle \nabla f(x; V), V \rangle - \langle \nabla f(x; W), W \rangle - \langle \nabla f(x; W), V - W \rangle \\
&= f(x; V) - f(x; W) - \langle \nabla f(x; W), V - W \rangle \\
&= -h_V(x) - \langle \nabla f(x; W), V - W \rangle,
\end{aligned}
$$

which matches the (negation of) functions considered in the function class $\mathcal{H}$ in Lemma B.3.

By the previous part, with $R_B := 0$, for any fixed $\|x\| \le 1$, with probability at least $1 - 3\delta$,

$$\sup_{h \in \mathcal{H}} |h(x)| \le \frac{6\rho R_V^{4/3} \ln(e/\delta)^{1/4}}{m^{1/6}} =: r_2.$$

Next, with probability at least $1 - \delta$, Lemma B.2 gives

$$\|W\|_2 \le \sqrt{m} + \sqrt{d} + \sqrt{2\ln(1/\delta)},$$

and thus for any $\|x - z\| \le \epsilon$, since the ReLU is 1-Lipschitz even when applied to vectors,

$$
\begin{aligned}
|h_V(x) - h_V(z)| &\le |f(x; V) - f(z; V)| + |f(x; W) - f(z; W)| \\
&\le \rho \|(V - W + W)(x - z)\| + \rho \|W(x - z)\| \\
&\le 2\rho\epsilon(R_V/2 + \sqrt{m} + \sqrt{d} + \sqrt{2\ln(1/\delta)}) =: r_1.
\end{aligned}
$$

Together, by Lemma B.3, choosing $\epsilon := 1/(dm)$ and $\mathcal{S}_0 := \mathbb{R}^{m \times d}$, with probability at least $1 - (1 + 3(md^{3/2})^d)\delta$,

$$\sup_{\|x\| \le 1} \sup_{h \in \mathcal{H}} |h(x)| \le r_2 + r_1 + 11 R_V \rho \left( \frac{\ln(edm/\delta)}{m} \right)^{1/4} \le \frac{25\rho R_V^{4/3} \sqrt{\ln(edm/\delta)}}{m^{1/6}}.$$

4. By the previous item, with probability at least $1 - (1 + 3(md^{3/2})^d)\delta$,

$$\sup_{\|W_i - W_0\| \le R_V} \sup_{\|x\| \le 1} \left| f^{(0)}(x; W_i) - f(x; W_i) \right| \le \tau_3.$$

Consequently, by Lemma B.1, for any $W_i$ with $\|W_i - W_0\| \le R_V$,

$$\mathcal{R}(W_i) = \mathbb{E}_{x,y} \ell(yf(x; W_i)) \le e^{\tau_3} \mathbb{E}_{x,y} \ell(yf^{(0)}(x; W_i)) = e^{\tau_3} \mathcal{R}^{(0)}(W_i).$$

$\square$

## B.2 Generalization proofs

As mentioned before, the usual hard part of such a proof is the Rademacher complexity estimate, but here it is easy: linear predictors, as this bound is applied after linear approximation. The difficult step is to control the range, which was presented before in Lemma B.6, which invokes the sphere control technique in Lemma B.3.

**Lemma B.8.** *Let $R_V \ge 1$ and $m \ge \ln(edm)$ be given. With probability at least $1 - 6\delta$,*

$$\sup_{\|V - W_0\| \le R_V} \mathcal{R}^{(0)}(V) - \widehat{\mathcal{R}}^{(0)}(V) \le \frac{80\rho R_V \left( d \ln(em^2 d^3/\delta) \right)^{3/2}}{\sqrt{n}}.$$

*Similarly, the negation of this bound holds with probability at least $1 - 6\delta$.*

*Proof.* This proof will use a constant $\delta_0$, chosen at the end. First note that the Rademacher complexity is as for linear predictors:

$$n\mathrm{Rad}\left(\left\{x \mapsto \langle \nabla f(x; W_0), V \rangle : \|V - W_0\| \le R_V \right\}\right) = \mathbb{E}_\epsilon \sup_{V \in \mathcal{V}} \sum_{k=1}^{n} \epsilon_k \langle \nabla f(x_k; W_0), V \rangle$$

$$= \mathbb{E}_\epsilon \sup_{V \in \mathcal{V}} \sum_{k=1}^{n} \epsilon_k \langle \nabla f(x_k; W_0), V - W_0 + W_0 \rangle$$

$$= \mathbb{E}_\epsilon \sup_{V \in \mathcal{V}} \sum_{k=1}^{n} \epsilon_k \langle \nabla f(x_k; W_0), V - W_0 \rangle$$

$$\le \|V - W_0\|_\mathrm{F} \sqrt{\sum_{k=1}^{n} \|\nabla f(x_k; W_0)\|^2}$$

$$\le \rho R_V \sqrt{n}.$$

Next, by Lemma B.6, with probability at least $1 - (1 + 3(md^2)^d)\delta_0$, the mappings $(x, y) \mapsto \ell(y f^{(0)}(x; V))$ are nonnegative, centered at $\ell(0)$, and vary by at most $18\rho R_V \ln(emd/\delta_0)$, thus take their amplitude to be $36\rho R_V \ln(emd/\delta_0)$ for simplicity. As such, since $\ell$ is 1-Lipschitz, by a standard Rademacher bound [Shalev-Shwartz and Ben-David, 2014], with additional failure probability at most $2\delta_0$,

$$\sup_{\|V - W_0\| \le R_V} \mathcal{R}^{(0)}(V) - \widehat{\mathcal{R}}^{(0)}(V) \le \frac{2\rho R_V}{\sqrt{n}} + \frac{108\rho R_V \ln(emd/\delta_0)\sqrt{\ln(1/\delta_0)}}{\sqrt{2n}}$$

$$\le \frac{80\rho R_V \ln(emd/\delta_0)^{3/2}}{\sqrt{n}},$$

and the bound is complete by noting the total failure probability was at most $(3 + 3(md^2)^d)\delta_0 \le 6(md^2)^d\delta_0$, and setting $\delta_0 := \delta/(md^2)^d$ and simplifying.

For the reverse inequality, it follows by negating every element in the loss class and repeating the proof. $\qquad\square$

### B.3 Optimization proofs

First, a smoothness inequality which fixes the feature mapping across a pair of iterates. This lemma doesn't seem to have appeared before, but is not necessarily an improvement, other than allowing slightly larger step sizes.

**Lemma B.9.** *For any step size $\eta \ge 0$,*

$$\eta(1 - \eta\rho^2/8)\|\nabla\widehat{\mathcal{R}}(W_i)\|^2 \le \widehat{\mathcal{R}}^{(i)}(W_i) - \widehat{\mathcal{R}}^{(i)}(W_{i+1}).$$

*If $\eta \le 8/\rho^2$, then $\widehat{\mathcal{R}}^{(i)}(W_{i+1}) \le \widehat{\mathcal{R}}^{(i)}(W_i)$, and any choice $\eta \le 4/\rho^2$ grants*

$$\frac{\eta}{2}\|\nabla\widehat{\mathcal{R}}(W_i)\|^2 \le \widehat{\mathcal{R}}^{(i)}(W_i) - \widehat{\mathcal{R}}^{(i)}(W_{i+1}).$$

*Proof.* For notational convenience, define $g_k(W) := y_k f(x_k; W)$ and $g_k^{(i)}(W) := y_k f^{(i)}(x_k; W)$, whereby $\nabla g_k(W) = y_k \nabla f(x_k; W)$. Since $\ell$ is $1/4$-smooth and since, for every example $(x_k, y_k)$, $\|\nabla f(x_k; V)\|^2 = \rho^2 \sum_{j=1}^{m} \|a_j \mathbb{1}[w_j^\mathsf{T} x_k \ge 0] x_k\|^2/m \le 1$, then

$$\ell(g_k^{(i)}(W_{i+1})) \le \ell(g_k^{(i)}(W_i)) + \ell'(g_k^{(i)}(W_i))(g_k^{(i)}(W_{i+1}) - g_k^{(i)}(W_i)) + \frac{1}{8}\left(g_k^{(i)}(W_{i+1}) - g_k^{(i)}(W_i)\right)^2$$

$$= \ell(g_k^{(i)}(W_i)) + \left\langle \ell'(g_k^{(i)}(W_i))\nabla g_k(W_i), W_{i+1} - W_i \right\rangle + \frac{1}{8}\left\langle \nabla g_k(W_i), W_{i+1} - W_i \right\rangle^2$$

$$= \ell(g_k^{(i)}(W_i)) - \eta\left\langle \ell'(g_k^{(i)}(W_i))\nabla g_k(W_i), \nabla\widehat{\mathcal{R}}(W_i) \right\rangle + \frac{1}{8}\left\langle \nabla g_k(W_i), \eta\nabla\widehat{\mathcal{R}}(W_i) \right\rangle^2$$

$$\le \ell(g_k^{(i)}(W_i)) - \eta\left\langle \ell'(g_k^{(i)}(W_i))\nabla g_k(W_i), \nabla\widehat{\mathcal{R}}(W_i) \right\rangle + \frac{\rho^2\eta^2}{8}\left\|\nabla\widehat{\mathcal{R}}(W_i)\right\|^2,$$

which after averaging over examples gives

$$\widehat{\mathcal{R}}^{(i)}(W_{i+1}) \leq \widehat{\mathcal{R}}^{(i)}(W_i) - \frac{\eta}{n}\sum_{k=1}^{n}\left\langle \ell'(g_k^{(i)}(W_i))\nabla g_k(W_i), \nabla\widehat{\mathcal{R}}(W_i)\right\rangle + \frac{\rho^2\eta^2}{8}\left\|\nabla\widehat{\mathcal{R}}(W_i)\right\|^2$$

$$= \widehat{\mathcal{R}}^{(i)}(W_i) - \eta(1 - \rho^2\eta/8)\left\|\nabla\widehat{\mathcal{R}}(W_i)\right\|^2,$$

which rearranges to give the first inequality. Lastly, note if $\eta \leq 4/\rho^2$, then $\eta\left(1 - \rho^2\eta/8\right) \geq \eta/2$. $\quad\square$

Next, the familiar regret inequality, making use of feature mappings induced by specific gradient descent iterates. Note that this inequality does not need to make any assumptions on nonlinearity and activation changes, though such effects must be controlled in the eventual application of this bound.

**Lemma B.10.** *For any step size $\eta \leq 4/\rho^2$, any $Z \in \mathbb{R}^{m \times d}$ and any $t$,*

$$\|W_t - Z\|^2 + 2\eta\sum_{i<t}\widehat{\mathcal{R}}^{(i)}(W_{i+1}) \leq \|W_0 - Z\|^2 + 2\eta\sum_{i<t}\widehat{\mathcal{R}}^{(i)}(Z).$$

*Proof.* As usual, using Lemma B.9,

$$\|W_{i+1} - Z\|^2 = \|W_i - Z\|^2 - 2\eta\left\langle \nabla\widehat{\mathcal{R}}(W_i), W_i - Z\right\rangle + \eta^2\|\nabla\widehat{\mathcal{R}}(W_i)\|^2$$

$$\leq \|W_i - Z\|^2 + 2\eta\left\langle \nabla\widehat{\mathcal{R}}(W_i), Z - W_i\right\rangle + 2\eta\left(\widehat{\mathcal{R}}^{(i)}(W_i) - \widehat{\mathcal{R}}^{(i)}(W_{i+1})\right),$$

where

$$\left\langle \nabla\widehat{\mathcal{R}}(W_i), Z - W_i\right\rangle = \frac{1}{n}\sum_k \ell'(y_k f(x_k; W_i))\left\langle y_k\nabla f(x_k; W_i), Z - W_i\right\rangle$$

$$= \frac{1}{n}\sum_k \ell'(y_k f(x_k; W_i))\left(y_k f^{(i)}(x_k; Z) - y_k f^{(i)}(x_k; W_i)\right)$$

$$\leq \frac{1}{n}\sum_k \left(\ell(y_k f^{(i)}(x_k; Z)) - \ell(y_k f^{(i)}(x_k; W_i))\right)$$

$$= \widehat{\mathcal{R}}^{(i)}(Z) - \widehat{\mathcal{R}}^{(i)}(W_i),$$

together giving

$$\|W_{i+1} - Z\|^2 \leq \|W_i - Z\|^2 + 2\eta\left(\widehat{\mathcal{R}}^{(i)}(Z) - \widehat{\mathcal{R}}^{(i)}(W_{i+1})\right),$$

which after telescoping and rearranging gives the final bound. $\quad\square$

Lastly, the proof of Lemma 2.2, the central optimization guarantee, which immediately yields the bulk of Theorem 1.1.

*Proof of Lemma 2.2.* The start of this proof establishes a few inequalities used throughout. By the second part of Lemma B.7, with probability at least $1 - 3n\delta$, for any iterations $(i, j)$ with $\|W_i - W_0\| \leq B$ and $\|W_j - W_0\| \leq B$,

$$\sup_{\|V - W_0\| \leq B} \frac{\widehat{\mathcal{R}}^{(i)}(V)}{\widehat{\mathcal{R}}^{(j)}(V)} \leq e^{\tau}. \tag{5}$$

Crucially, eq. (5) holds with $V := Z$, since $B \geq R_Z$ by definition. Additionally, by Lemma B.10, the following inequality holds *unconditionally* for every $j \leq t$:

$$\|W_j - Z\|^2 + 2\eta\sum_{i<j}\widehat{\mathcal{R}}^{(i)}(W_{i+1}) \leq \|W_0 - Z\|^2 + 2\eta\sum_{i<j}\widehat{\mathcal{R}}^{(i)}(Z). \tag{6}$$

The remainder of the proof is broken into three parts, for the three separate guarantees:

$$\|W_{\leq t} - W_0\| \leq B \qquad\qquad\qquad\qquad\qquad\qquad\qquad \text{(norm),} \quad (7)$$

$$\widehat{\mathcal{R}}(W_{\leq t}) \leq e^{2\tau}\widehat{\mathcal{R}}^{(0)}(Z) + e^{\tau}(\rho R_Z)^2\epsilon_{\text{gd}} \qquad\qquad\quad \text{(empirical risk),} \quad (8)$$

$$\mathcal{R}(W_{\leq t}) \leq e^{4\tau}\mathcal{R}^{(0)}(Z) + e^{3\tau}(\rho R_Z)^2\epsilon_{\text{gd}} + e^{4\tau}\rho(B + R_Z)\tau_n \qquad \text{(risk).} \quad (9)$$

**Norm guarantee (cf. eq. (7)).** There are two cases to consider: $B = R_{\text{gd}}$, or $B < R_{\text{gd}}$. If $B = R_{\text{gd}}$, the claim follows by the definition of $W_{\leq t}$.

Now suppose $B < R_{\text{gd}}$, meaning $B = 3R_Z + 2e\sqrt{\eta t \widehat{\mathcal{R}}^{(0)}(Z)}$. It will now be argued via contradiction that $\max_{i \leq t} \|W_i - W_0\| \leq B$. Assume contradictorily the claim does not hold, and let $s \leq t$ be the earliest violation. But that means the claim holds for all $i < s$, which also means, combining eq. (5) (which must hold for all $i < s$) and eq. (6) and using $\tau \leq 2$ and $\ell \geq 0$,

$$
\begin{aligned}
B^2 < \|W_s - W_0\|^2 &\leq 2\|W_s - Z\|^2 + 2\|Z - W_0\|^2 \\
&\leq 2\|W_s - Z\|^2 + 4\eta \sum_{i<s} \widehat{\mathcal{R}}^{(i)}(W_{i+1}) + 2\|Z - W_0\|^2 \\
&\leq 4\|W_0 - Z\|^2 + 4\eta \sum_{i<s} \widehat{\mathcal{R}}^{(i)}(Z) \\
&\leq 4\|W_0 - Z\|^2 + 4\eta t e^2 \widehat{\mathcal{R}}^{(0)}(Z) \\
&\leq \left( 2\|W_0 - Z\| + 2e\sqrt{\eta t \widehat{\mathcal{R}}^{(0)}(Z)} \right)^2 \leq B^2,
\end{aligned}
$$

a contradiction.

**Empirical risk guarantee (cf. eq. (8)).** Now let $T$ denote the earliest time when $\|W_i - W_0\| > 2R_Z$, or $T = \infty$ if this situation never occurs. Note that for any $i < T$,

$$\|W_i - W_0\| \leq 2R_Z \leq B,$$

and even for $W_T$,

$$\|W_T - W_0\| \leq \|W_{T-1} - W_0\| + \eta \|\nabla \widehat{\mathcal{R}}(W_{T-1})\| \leq 2R_Z + \eta \rho \leq B;$$

as such, eq. (5) holds for all $W_i$ with $i \leq T$, including the edge case $W_T$. The remainder of the proof divides into two cases: either $T > t$ (which includes the situation $T = \infty$), or $T \leq t$.

If $T \leq t$, by the triangle inequality,

$$2\|Z - W_0\| < \|W_T - W_0\| \leq \|Z - W_T\| + \|Z - W_0\|,$$

which rearranges to give $\|Z - W_0\| < \|Z - W_T\|$, and thus, by eq. (6),

$$
\begin{aligned}
\|Z - W_0\|^2 + 2\eta \sum_{i<T} e^{-\tau} \widehat{\mathcal{R}}(W_{i+1}) &< \|W_T - Z\|^2 + 2\eta \sum_{i<T} \widehat{\mathcal{R}}^{(i)}(W_{i+1}) \\
&\leq \|Z - W_0\|^2 + 2\eta \sum_{i<T} \widehat{\mathcal{R}}^{(i)}(Z) \\
&\leq \|Z - W_0\|^2 + 2\eta \sum_{i<T} e^{\tau} \widehat{\mathcal{R}}^{(0)}(Z),
\end{aligned}
$$

which after canceling from both sides and using the definition of $W_{\leq t}$,

$$\widehat{\mathcal{R}}(W_{\leq t}) \leq \min_{i<T} \widehat{\mathcal{R}}(W_i) \leq \frac{1}{T} \sum_{i<T} \widehat{\mathcal{R}}(W_i) \leq e^{2\tau} \widehat{\mathcal{R}}^{(0)}(Z),$$

establishing eq. (8) when $T \leq t$.

If $T > t$, the proof is simpler: since $\max_{i \leq t} \|W_i - W_0\| \leq 2R_Z \leq B$, then eq. (5) holds for all $W_i$ with $i \leq t$, and thus by eq. (6) and the definition of $W_{\leq t}$,

$$
\begin{aligned}
2\eta e^{-\tau} \sum_{i<t} \widehat{\mathcal{R}}(W_{i+1}) \leq 2\eta \sum_{i<t} \widehat{\mathcal{R}}^{(i)}(W_{i+1}) &\leq \|W_t - Z\|^2 + 2\eta \sum_{i<t} \widehat{\mathcal{R}}^{(i)}(W_{i+1}) \\
&\leq \|W_0 - Z\|^2 + 2\eta \sum_{i<t} \widehat{\mathcal{R}}^{(i)}(Z) \\
&\leq \|W_0 - Z\|^2 + 2t\eta e^{\tau} \widehat{\mathcal{R}}^{(0)}(Z),
\end{aligned}
$$

which after rearranging and using the definition of $W_{\leq t}$ gives

$$\widehat{\mathcal{R}}(W_{\leq t}) \leq \frac{1}{t} \sum_{i<t} \widehat{\mathcal{R}}(W_{i+1}) \leq e^{2\tau} \widehat{\mathcal{R}}^{(0)}(Z) + \frac{e^{\tau}\|W_0 - Z\|^2}{2t\eta} \leq e^{2\tau} \widehat{\mathcal{R}}^{(0)}(Z) + e^{\tau}(\rho R_Z)^2 \epsilon_{\text{gd}},$$

completing the proof of eq. (8).

**Risk guarantee (cf. eq. (9)).** By Lemma B.8 applied once with radius $B$ and once with radius $R_Z$, with probability at least $1 - 12\delta$

$$\mathcal{R}^{(0)}(W_{\leq t}) \leq \widehat{\mathcal{R}}^{(0)}(W_{\leq t}) + \rho B \tau_n, \qquad \widehat{\mathcal{R}}^{(0)}(Z) \leq \mathcal{R}^{(0)}(Z) + \rho R_Z \tau_n.$$

Moreover, by the last part of Lemma B.7 applied with radius $B$ with probability at least $1 - 4\delta$,

$$\mathcal{R}(W_{\leq t}) \leq e^{\tau} \mathcal{R}^{(0)}(W_{\leq t}).$$

Combining all these inequalities with the empirical risk guarantee,

$$
\begin{aligned}
\mathcal{R}(W_{\leq t}) &\leq e^{\tau} \mathcal{R}^{(0)}(W_{\leq t}) \\
&\leq e^{\tau} \widehat{\mathcal{R}}^{(0)}(W_{\leq t}) + e^{\tau} \rho B \tau_n \\
&\leq e^{2\tau} \widehat{\mathcal{R}}(W_{\leq t}) + e^{\tau} \rho B \tau_n \\
&\leq e^{4\tau} \widehat{\mathcal{R}}^{(0)}(Z) + e^{3\tau}(\rho R_Z)^2 \epsilon_{\mathrm{gd}} + e^{\tau} \rho B \tau_n \\
&\leq e^{4\tau} \mathcal{R}^{(0)}(Z) + e^{3\tau}(\rho R_Z)^2 \epsilon_{\mathrm{gd}} + \rho \left( e^{\tau} B + e^{4\tau} R_Z \right) \tau_n,
\end{aligned}
$$

thus establishing eq. (9) and completing the proof. $\qquad\square$

## B.4 Approximation proofs

First, the lemma and proof that we can sample from $\overline{U}_\infty$; as the gap is over the risk, the proof uses the technique in Lemma B.3 to control all points on the sphere. This proof also makes crucial use of the arccos bound in Lemma B.4.

**Lemma B.11.** *Let $\overline{U}_\infty$ be given with $R := \sup_{v \in \mathbb{R}^d} \|\overline{U}_\infty(v)\|$, and suppose $m \geq \ln(emd)$. With probability at least $1 - 6\delta$,*

$$\mathcal{R}^{(0)}(\overline{U}) \leq e^{\tau} \mathcal{R}(\overline{U}_\infty), \qquad \text{where } \tau \leq 6\rho d \ln(emd^2/\delta) + \frac{20R\sqrt{d \ln(em^2 d^3/\delta)}}{m^{1/4}}.$$

*Proof of Lemma B.11.* Throughout this proof, the subscript will be dropped and simply $W := W_0$, with rows $(w_j^{\mathsf{T}})_{j=1}^m$.

The bound on $\mathcal{R}^{(0)}(\overline{U}) - \mathcal{R}(\overline{U}_\infty)$ follows by showing that with probability at least $1 - 6\delta$,

$$\sup_{\|x\| \leq 1} \left| f(x; \overline{U}_\infty) - f^{(0)}(x; \overline{U}) \right| \leq \tau,$$

and then as usual applying Lemma B.1 and taking an expectation to obtain a bound between $\mathcal{R}^{(0)}(\overline{U})$ and $\mathcal{R}(\overline{U}_\infty)$. Meanwhile, this intermediate bound is first established for any fixed $x \in \mathbb{R}^d$, and then general $\|x\| \leq 1$ are handled via Lemma B.3.

Fix an example $x \in \mathbb{R}^d$ and failure probability $\delta_0$ to be determined later when Lemma B.3 is invoked. To first calculate the expected difference, note by definition of $\overline{U}$ that

$$
\begin{aligned}
\mathbb{E}\left\langle \nabla f(x; W), \overline{U} - W \right\rangle &= \mathbb{E}\frac{\rho}{\sqrt{m}} \sum_{j=1}^m a_j \left\langle \overline{u}_j - w_j, x\mathbb{1}[w_j^{\mathsf{T}} x \geq 0] \right\rangle \\
&= \frac{1}{m} \sum_{j=1}^m \mathbb{E}\left\langle \overline{U}_\infty(w_j), x\mathbb{1}[w_j^{\mathsf{T}} x \geq 0] \right\rangle \\
&= f(x; \overline{U}_\infty),
\end{aligned}
$$

whereas

$$\mathbb{E}\left\langle \nabla f(x; W), W \right\rangle = \mathbb{E}_a \sum_{j=1}^m a_j \mathbb{E}_{w_j} \sigma_{\mathrm{r}}(w_j^{\mathsf{T}} x) = 0,$$

thus

$$\mathbb{E} f^{(0)}(x; \overline{U}) = \mathbb{E}\left( f^{(0)}(x; \overline{U} - W) + f^{(0)}(x; W) \right) = f(x; \overline{U}_\infty).$$

Controlling the deviations (still for this fixed $x$) will also consider the terms separately. The term $f^{(0)}(x; \overline{U} - W)$ will use McDiarmid's inequality; to verify the bounded differences property, consider pairs $(a, W)$ and $(a', W')$ which differ in only one element $(a'_j, w'_j)$, which also defines pairs $\overline{U}$ and $\overline{U}'$ differing in just one $j$, meaning the vectors $\overline{u}_j$ and $\overline{u}'_j$; by Cauchy-Schwarz and the definition of $R$,

$$
\begin{aligned}
&\left| \langle \nabla f(W), U - W \rangle - \langle \nabla f(W'), U' - W' \rangle \right| \\
&= \left| \frac{\rho}{\sqrt{m}} a_j \langle \overline{u}_j - w_j, x \mathbb{1}[w_j^\mathsf{T} x_j \geq 0] \rangle - \frac{\rho}{\sqrt{m}} a'_j \langle \overline{u}'_j - w'_j, x \mathbb{1}[(w'_j)^\mathsf{T} x_j \geq 0] \rangle \right| \\
&= \frac{1}{m} \left| a_j^2 \langle \overline{U}_\infty(w_j), x \mathbb{1}[w_j^\mathsf{T} x_j \geq 0] \rangle - (a'_j)^2 \langle \overline{U}_\infty(w'_j), x \mathbb{1}[(w'_j)^\mathsf{T} x_j \geq 0] \rangle \right| \\
&\leq \frac{2R\|x\|}{m}.
\end{aligned}
$$

Thus, by McDiarmid's inequality, with probability at least $1 - 2\delta_0$,

$$
\left| f^{(0)}(x; \overline{U}) - f(x; \overline{U}_\infty) \right| = \left| f^{(0)}(x; \overline{U}) - \mathbb{E}_{a,W} f^{(0)}(x; \overline{U}) \right| \leq \sqrt{\frac{2R^2 \|x\|^2 \ln(1/\delta_0)}{m}}.
$$

Meanwhile, the term $f^{(0)}(x; W)$ is explicitly controlled in in the first part of Lemma B.6: with probability at least $1 - 3\delta_0$,

$$
|f^{(0)}(x; W)| \leq 2\rho \|x\| \ln(1/\delta_0).
$$

Together, with probability at least $1 - 5\delta_0$,

$$
\left| f^{(0)}(x; \overline{U}) - f(x; \overline{U}_\infty) \right| \leq 2\rho \ln(1/\delta_0) + R \sqrt{\frac{2 \ln(1/\delta_0)}{m}} =: r_2.
$$

Controlling the behavior for all $\|x\| \leq 1$ simultaneously now relies upon Lemma B.3, but invoked to control a single matrix, namely choosing $\mathcal{S}_0 := \{\overline{U}\}$, and radius $R_V := R/\rho \geq \|\overline{U} - W\|$. For the sake of applying Lemma B.3, define for any $V \in \mathbb{R}^{m \times d}$ the mapping

$$
h_V(x) := f(x; W) - f(x; \overline{U}_\infty),
$$

which has no dependence on $V$, and note a corresponding function $h \in \mathcal{H}$ as defined in Lemma B.3 has the form

$$
h(x) = f(x; W) - f(x; \overline{U}_\infty) + \langle \nabla f(x; W), V - W \rangle = \langle \nabla f(x; W), V \rangle - f(x; \overline{U}_\infty);
$$

since $\mathcal{S}_0 = \{\overline{U}\}$, we only need to check the conditions of Lemma B.3 for $V = \overline{U}$. As above, for any fixed $\|x\| \leq 1$, with probability at least $1 - 5\delta_0$, $|h(x)| \leq r_2$. To invoke Lemma B.3, the restricted continuity property must be established. Specifically, let $\|x - z\| \leq \epsilon$ be given, with $\epsilon > 0$ determined later. Writing

$$
\left| h_V(x) - h_V(z) \right| \leq \left| f(x; W) - f(z; W) \right| + \left| f(x; \overline{U}_\infty) - f(z; \overline{U}_\infty) \right|,
$$

it suffices to check the restricted continuity property in both terms separately. For the first term, by Lemma B.2, with probability at least $1 - \delta_0$,

$$
\|W\|_2 \leq \sqrt{m} + \sqrt{d} + \sqrt{2 \ln(1/\delta_0)},
$$

whereby the 1-Lipschitz property of the ReLU over vectors gives

$$
\left| f(x; W) - f(z; W) \right| \leq \rho \|\sigma_\mathrm{r}(Wx) - \sigma_\mathrm{r}(Wz)\| \leq \rho \|W(x-z)\| \leq \rho \left( \sqrt{m} + \sqrt{d} + \sqrt{2 \ln(1/\delta_0)} \right) \epsilon.
$$

For the other term, first note by a standard Gaussian calculation that

$$
\begin{aligned}
\left| f(z; \overline{U}_\infty) - f(x; \overline{U}_\infty) \right| &= \left| \int \langle \overline{U}_\infty(v), z \mathbb{1}[v^\mathsf{T} z \geq 0] - x \mathbb{1}[v^\mathsf{T} x \geq 0] \rangle \, \mathrm{d}\mathcal{N}(v) \right| \\
&\leq R \int \left\| z \mathbb{1}[v^\mathsf{T} z \geq 0] - x \mathbb{1}[v^\mathsf{T} x \geq 0] \right\| \, \mathrm{d}\mathcal{N}(v) \\
&\leq R \|z - x\| \Pr_{v \sim \mathcal{N}} \left[ \mathbb{1}[v^\mathsf{T} z \geq 0] = \mathbb{1}[v^\mathsf{T} x \geq 0] \right] \\
&\quad + R(\|x\| + \|z\|) \Pr_{v \sim \mathcal{N}} \left[ \mathbb{1}[v^\mathsf{T} z \geq 0] \neq \mathbb{1}[v^\mathsf{T} x \geq 0] \right] \\
&\leq R \|z - x\| + R(\|x\| + \|z\|) \frac{2 \arccos(\langle x/\|x\|, z/\|z\| \rangle)}{2\pi}.
\end{aligned}
$$

If $\|x\| \le 2\epsilon$, then $z \le 3\epsilon$, and the last term can be upper bounded as $5R\epsilon$. On the other hand, if $\|x\| > 2\epsilon$, whereby $\|x\| + \|z\| \le 2\|x\| + \epsilon \le 3\|x\|$, then Lemma B.4 implies

$$R(\|x\| + \|z\|) \frac{2\arccos(\langle x/\|x\|, z/\|z\|\rangle)}{2\pi} \le R(\|x\| + \|z\|) \frac{\epsilon\sqrt{8}}{\|x\|\pi} \le 3R\epsilon,$$

Thus, by Lemma B.3 with radius $R_V := R/\rho$ and filter set $\mathcal{S}_0 := \{\overline{U}\}$ as above, and additionally choosing $\epsilon := 1/(dm)$, with overall probability at least $1 - (1 + 5(\sqrt{d}/\epsilon)^d)\delta_0$,

$$\sup_{\|x\| \le 1} \left| f^{(0)}(x; \overline{U}) - f(x; \overline{U}_\infty) \right| \le 2\rho \ln(e/\delta_0) + R\sqrt{\frac{2\ln(e/\delta_0)}{m}}$$

$$+ \epsilon\rho\left(\sqrt{m} + \sqrt{d} + \sqrt{2\ln(e/\delta_0)}\right) + (1+5)R\epsilon$$

$$+ 11R_V\rho\left(\frac{\ln(edm/\delta_0)}{m}\right)^{1/4}$$

$$\le 6\rho\ln(e/\delta_0) + 20R_V\rho\left(\frac{\ln(edm/\delta_0)}{m}\right)^{1/4},$$

and the final bound comes via the choice $\delta_0 := \delta/(md^2)^d$. $\qquad\square$

The next result establishes that for any $p_y$, there exists a conditional probability model defined by $\overline{U}_\infty$ which is arbitrarily close, which is one of the keys to the consistency proof (cf. Corollary 2.3). As discussed briefly in Remark 1.1, this construction requires a bias term, which is simulated by replacing the input $x \in \mathbb{R}^d$ with $(x, 1)/\sqrt{2} \in \mathbb{R}^{d+1}$, and otherwise proceeding without modification.

**Lemma B.12.** *Suppose $\mu_x$ and $p_y$ are Borel measurable, and $\mu_x$ is supported on $\|x\| \le 1$. Given any $x \in \mathbb{R}^d$, let $\tilde{x} := (x, 1)/\sqrt{2} \in \mathbb{R}^{d+1}$ denote the vector obtained by appending the constant 1. Then for any $\epsilon > 0$, there exist infinite-width weights $\overline{U}_\infty : \mathbb{R}^{d+1} \to \mathbb{R}^{d+1}$ satisfying $R := \sup_{\tilde{v} \in \mathbb{R}^{d+1}} \overline{U}_\infty(\tilde{v}) < \infty$ and*

$$\mathcal{R}(\overline{U}_\infty) \le \overline{\mathcal{R}} + \epsilon.$$

*Proof.* Throughout this proof, define $\tau := \min\{\epsilon/4, 1/2\}$.

As is standard in the theory of classification calibration [Zhang, 2004, Bartlett et al., 2006], for the logistic loss, the optimal population risk is achieved by a measurable function $\bar{f} : \mathbb{R} \to \mathbb{R}$ which satisfies

$$\bar{f}(x) := \underset{r \in \mathbb{R} \cup \pm\infty}{\arg\min}\, p_y(x)\ell(r) + (1 - p_y(x))\ell(-r) = \phi^{-1}(p_y(x)) = \ln\frac{p_y(x)}{1 - p_y(x)} \qquad \mu_x\text{-a.e. } x,$$

which may take on the values $\pm\infty$. To avoid these $\pm\infty$, define a clamping of $p_y$ as

$$p_1(x) := \max\{\tau, \min\{1 - \tau, p_y(x)\}\},$$

and clamped logits $f_1(x) := \phi^{-1}(p_1(x))$ (which now is bounded). As is again usual in the literature on classification calibration [Zhang, 2004, Bartlett et al., 2006],

$$\mathcal{R}(f_1) - \overline{\mathcal{R}} = \int\left(p_y(x)\ln\frac{p_y(x)}{p_1(x)} + (1 - p_y(x))\ln\frac{1 - p_y(x)}{1 - p_1(x)}\right)\mathrm{d}\mu_x(x)$$

$$= \int_{p_y(x) \in [0, \tau)}\left(p_y(x)\ln\frac{p_y(x)}{\tau} + (1 - p_y(x))\ln\frac{1 - p_y(x)}{1 - \tau}\right)\mathrm{d}\mu_x(x)$$

$$+ \int_{p_y(x) \in (1-\tau, 1]}\left(p_y(x)\ln\frac{p_y(x)}{1 - \tau} + (1 - p_y(x))\ln\frac{1 - p_y(x)}{\tau}\right)\mathrm{d}\mu_x(x)$$

$$\le \frac{\tau}{1 - \tau} \le 2\tau.$$

Since $p_1$ is Borel measurable (due to Borel measurability of $p_y$), then $f_1$ is Borel measurable (since $\phi^{-1}$ is continuous along $[\tau, 1 - \tau]$), and therefore we may apply Lusin's Theorem [Folland, 1999, Theorem 7.10]: there exists a continuous function $g$ and a set $S$ satisfying

$$|g| \le |f_1| \le \sup_x |f_1(x)| < \infty, \qquad g_{|S} = (f_1)_{|S}, \qquad \mu_x(S^c) \le \frac{\tau}{\ell(0) + \sup_x |f_1(x)|},$$

whereby since $\ell$ is 1-Lipschitz,

$$\mathcal{R}(g) - \mathcal{R}(f_1) \leq \int \mathbb{1}[x \in S^c]\ell(-yg(x))\,\mathrm{d}\mu(x,y)$$

$$\leq \int \mathbb{1}[x \in S^c]\ell(|g(x)|)\,\mathrm{d}\mu_x(x)$$

$$\leq \mu_x(S^c)(\ell(0) + \sup_x |g(x)|)$$

$$\leq \tau.$$

Since $g$ is continuous, it is uniformly continuous over $\|x\| \leq 1$, and thus there exists a $\delta > 0$ so that the modulus of continuity $\omega_g(\delta)$ at scale $\delta$ is at most $\tau$, meaning

$$\sup_{\|x-x'\| \leq \delta} |g(x) - g(x')| \leq \omega_g(\delta) \leq \tau.$$

By results in neural network universal approximation [Ji et al., 2020b, Theorem 4.3], there exists infinite-width weights $\overline{U}_\infty : \mathbb{R}^{d+1} \to \mathbb{R}^{d+1}$ satisfying $R := \sup_{\tilde{x}} \|\overline{U}_\infty(\tilde{x})\| < \infty$ and

$$\sup_{\|x\| \leq 1} \left| f(\tilde{x}; \overline{U}_\infty) - g(x) \right| \leq \omega_g(\delta) \leq \tau,$$

which again by the 1-Lipschitz property of $\ell$ means $\mathcal{R}(\overline{U}_\infty) - \mathcal{R}(g) \leq \tau$. Combining all these pieces,

$$\mathcal{R}(\overline{U}_\infty) - \overline{\mathcal{R}} = \left[\mathcal{R}(\overline{U}_\infty) - \mathcal{R}(g)\right] + \left[\mathcal{R}(g) - \mathcal{R}(f_1)\right] + \left[\mathcal{R}(f_1) - \overline{\mathcal{R}}\right] \leq \tau + \tau + 2\tau \leq \epsilon,$$

as desired. $\qquad\square$

## B.5 Proofs of main results: Theorem 1.1 and Corollary 2.3

The proof of Theorem 1.1 and a precise restatement are as follows. This restatement has fully explicit constants, and is invoked in the proof of Corollary 2.3 to ease sanity-checking.

**Theorem B.13** (Refined restatement of Theorem 1.1). *Let temperature $\rho > 0$ and reference model $\overline{U}_\infty$ be given with $R := \max\{4, \rho, \sup_v \|\overline{U}_\infty(v)\|\} < \infty$, and define a corresponding conditional model $\phi_\infty(x) := \phi(f(x; \overline{U}_\infty))$. Let optimization accuracy $\epsilon_{\mathrm{gd}}$ and radius $R_{\mathrm{gd}} \geq R/\rho$ be given, define effective radius $B := \min\left\{R_{\mathrm{gd}}, \frac{3R}{\rho} + \frac{4e}{\rho}\sqrt{t}\sqrt{e^{\tau_0}\mathcal{R}(\overline{U}_\infty) + R\tau_n}\right\}$, where generalization error $\tau_n$ and additionally linearization error $\tau_1$ and sampling error $\tau_0$ are defined as*

$$\tau_n := \frac{80\left(d\ln(em^2d^3/\delta)\right)^{3/2}}{\sqrt{n}},$$

$$\tau_1 := \frac{100\rho B^{4/3}\sqrt{d\ln(enm^2d^3/\delta)}}{m^{1/6}},$$

$$\tau_0 := 6\rho d\ln(emd^2/\delta) + \frac{20R\sqrt{d\ln(em^2d^3/\delta)}}{m^{1/4}},$$

*where it is assumed $\tau_1 \leq 2$ and $m \geq \ln(emd)$. Choose step size $\eta := 4/\rho^2$, and run gradient descent for $t := 1/(8\epsilon_{\mathrm{gd}})$ iterations, selecting iterate $W_{\leq t} := \arg\min\{\widehat{\mathcal{R}}(W_i) : i \leq t, \|W_i - W_0\| \leq R_{\mathrm{gd}}\}$ with simultaneously small norm and empirical risk. Then, with probability at least $1 - 25\delta$,*

$$
\begin{aligned}
&\mathcal{R}(W_{\leq t}) - \overline{\mathcal{R}} && \textit{(logistic error)}\\
\leq\ & \mathcal{K}_{\mathrm{bin}}(p_y, \phi_\infty) + \left(e^{\tau_1 + \tau_0} - 1\right)\mathcal{R}(\overline{U}_\infty) && \textit{(reference model error)}\\
+\ & e^{\tau_1}R^2\epsilon_{\mathrm{gd}} && \textit{(optimization error)}\\
+\ & e^{\tau_1}(\rho B + R)\tau_n && \textit{(generalization error),}
\end{aligned}
$$

*where the classification and calibration errors satisfy*

$$
\begin{aligned}
&\mathcal{R}(W_{\leq t}) - \overline{\mathcal{R}} && \textit{(logistic error)}\\
\geq\ & 2\int \left(\phi(f(x; W_{\leq t})) - p_y\right)^2 \mathrm{d}\mu_x(x) && \textit{(calibration error)}\\
\geq\ & \frac{1}{2}\left(\mathcal{R}_{\mathrm{z}}(W_{\leq t}) - \overline{\mathcal{R}}_{\mathrm{z}}\right)^2 && \textit{(classification error).}
\end{aligned}
$$

*Lastly, for any $\epsilon > 0$, there exists $\overline{U}_\infty^{(\epsilon)}$ with $\sup_v \|\overline{U}_\infty^{(\epsilon)}(v)\| < \infty$ and whose conditional model $\phi_\infty^{(\epsilon)}(x) := \phi(f((x,1)/\sqrt{2}; \overline{U}_\infty^{(\epsilon)}))$ satisfies $\mathcal{K}_{\mathrm{bin}}(p_y, \phi_\infty^{(\epsilon)}) \le \epsilon$.*

*Proof of Theorem 1.1 and simultaneously Theorem B.13.* This proof focuses on the first inequality, upper bounding $\mathcal{R}(W_{\le t}) - \overline{\mathcal{R}}$; for the other two statements, the chain of inequalities with other error metrics are from Lemma B.1, and the approximation of arbitrary Borel measurable $p_y$ is from Lemma B.12. (The only difference between Theorem B.13 here and Theorem 1.1 in the body is that the "$\widetilde{\mathcal{O}}$" hides constants and $\ln(m)$ and $\ln(d)$ (but not $\ln(n)$).

Returning to the first inequality, let $\overline{U}$ be the canonical sample of $\overline{U}_\infty$ as in eq. (1), where $\|\overline{U} - W_0\| \le R/\rho$ by construction. By Lemma B.11, with probability at least $1 - 6\delta$, then $\mathcal{R}^{(0)}(\overline{U}) \le e^{\tau_0} \mathcal{R}(\overline{U}_\infty)$, where $\tau_0$ is as in the statement (cf. Theorem B.13).

Next instantiate Lemma 2.2 with reference matrix $Z = \overline{U}$ and $R_Z := R/\rho$, whereby the definition of $R$ gives $R_Z \ge \{1, \eta\rho, \|\overline{U} - W_0\|\}$ as needed; as such, ignoring an additional failure probability at most $19\delta$, setting $\tau := \tau_1/4$ in the invocation, and lastly subtracting $\overline{\mathcal{R}}$ from both sides,

$$\mathcal{R}(W_{\le t}) - \overline{\mathcal{R}} \le e^{\tau_1} \mathcal{R}^{(0)}(\overline{U}) + e^{\tau_1}(\rho R_Z)^2 \epsilon_{\mathrm{gd}} + e^{\tau_1}(\rho B + \rho R_Z)\tau_n - \overline{\mathcal{R}}$$
$$\le \left(e^{\tau_1 + \tau_0} - 1\right) \mathcal{R}(\overline{U}_\infty) + \mathcal{K}_{\mathrm{bin}}(p_y, \phi_\infty) + e^{\tau_1} R^2 \epsilon_{\mathrm{gd}} + e^{\tau_1}(\rho B + R)\tau_n.$$

This invocation of Lemma 2.2 also guarantees $\widehat{\mathcal{R}}^{(0)}(\overline{U}) \le \mathcal{R}^{(0)}(\overline{U}) + R\tau_n$ which together with the earlier inequality $\mathcal{R}^{(0)}(\overline{U}) \le e^{\tau_0} \mathcal{R}(\overline{U}_\infty)$ provides the form of $B$ used in the statement (this $B$ upper bounds the one defined in Lemma 2.2, which is fine since it only relaxes the guarantees provided there). $\qquad\square$

Making use of Theorem B.13, the proof of the consistency statement, Corollary 2.3, is as follows. Note that we are always working with bias-augmented inputs within this statement and its proof; e.g., $\widehat{W}_n \in \mathbb{R}^{m^{(n)} \times (d+1)}$.

*Proof of Corollary 2.3.* Let $\epsilon > 0$ be arbitrary, and define the event

$$E_n := \left[ \mathcal{R}(\widehat{W}_n) \ge \overline{\mathcal{R}} + \epsilon \right].$$

Following a standard scheme for consistency proofs [Schapire and Freund, 2012, Corollary 12.3], it suffices, thanks to the Borel-Cantelli lemma, to prove

$$\sum_{n \ge 1} \Pr[E_n] < \infty; \tag{10}$$

that is to say, by the Borel-Cantelli lemma, eq. (10) implies $\limsup_{n \to \infty} \mathcal{R}(\widehat{W}_n) - \overline{\mathcal{R}} \le \epsilon$ almost surely, and since $\mathcal{R}(\widehat{W}_n) \ge \overline{\mathcal{R}}$ and since $\epsilon > 0$ was arbitrary, it follows that $\mathcal{R}(\widehat{W}_n) \to \overline{\mathcal{R}}$ almost surely. Moreover, by Lemma B.1, for each $n$ there are the inequalities

$$\frac{1}{2} \left( \mathcal{R}_z(\widehat{W}_n) - \overline{\mathcal{R}}_z \right)^2 \le 2 \int (\widehat{\phi}_n(x) - p_y(x))^2 \, d\mu_x(x) \le \mathcal{R}(\widehat{W}_n) - \overline{\mathcal{R}},$$

thus $\mathcal{R}(\widehat{W}_n) \to \overline{\mathcal{R}}$ also implies $\widehat{\phi}_n \to p_y$ in $L_2(\mu_x)$ almost surely, and $\mathcal{R}_z(\widehat{W}_n) \to \overline{\mathcal{R}}_z$ almost surely.

To establish eq. (10), first use the last part of Theorem B.13 to fix a $\overline{U}_\infty$ with $\mathcal{K}_{\mathrm{bin}}(p_y, \widehat{\phi}_n) \le \epsilon/2$, and define $R := \sup_v \|\overline{U}_\infty(v)\| < \infty$. To bound $\Pr[E_n]$, instantiate Theorem B.13 for every $n$ with reference model $\overline{U}_\infty$ and corresponding $R < \infty$, and failure probability $\delta^{(n)} := 1/n^2$, and optimization radius $R_{\mathrm{gd}} = \infty$, meaning a corresponding effective radius given by Theorem B.13 as

$$B^{(n)} = \frac{1}{\rho^{(n)}} \left( 3R + 4e\sqrt{t^{(n)}} \sqrt{e^{\tau_0^{(n)}} \mathcal{R}(\overline{U}_\infty) + R\tau_n} \right).$$

Inspecting all the terms in Theorem B.13, it will now be argued that while the term $\mathcal{K}_{\mathrm{bin}}(p_y, \widehat{\phi}_n)$ stays level and is at most $\epsilon/2$ independent of $n$, all other terms go to 0. Returning to $B^{(n)}$, since

$\tau_n = \widetilde{\mathcal{O}}(1/\sqrt{n})$ and $\tau_0^{(n)} \to 0$ (which will be shown later), then $B^{(n)} = \widetilde{\mathcal{O}}(\sqrt{t^{(n)}}/\rho^{(n)})$, whereby

$$\tau_1^{(n)} = \widetilde{\mathcal{O}}\left(\frac{\rho^{(n)}(B^{(n)})^{4/3}}{(m^{(n)})^{1/6}}\right) = \widetilde{\mathcal{O}}\left(\frac{(t^{(n)})^{2/3}}{(m^{(n)})^{1/6}(\rho^{(n)})^{1/3}}\right)$$

$$= \widetilde{\mathcal{O}}\left(\frac{(t^{(n)})^{2/3}}{(m^{(n)})^{1/8}}\right) = \widetilde{\mathcal{O}}\left(\frac{n^{\frac{2}{3}(1-\xi)}}{n^{\frac{5}{3}(1-\xi)}}\right) = \widetilde{\mathcal{O}}\left(n^{\xi-1}\right) \to 0.$$

Next,

$$\tau_0^{(n)} = \widetilde{\mathcal{O}}\left(\rho^{(n)} + \frac{1}{(m^{(n)})^{1/4}}\right) = \widetilde{\mathcal{O}}\left(n^{\frac{5}{3}(\xi-1)} + n^{\frac{10}{3}(\xi-1)}\right) \to 0,$$

which together with the asymptotics of $\tau_1^{(n)}$ gives $\exp(\tau_0^{(n)} + \tau_1^{(n)}) - 1 \to 0$ and $\exp(\tau_1^{(n)})R^2\epsilon_{\mathrm{gd}}^{(n)} \to 0$. The final term to consider is

$$\exp(\tau_1^{(n)})\rho^{(n)}B^{(n)}\tau_n = \widetilde{\mathcal{O}}\left(\sqrt{\frac{t^{(n)}}{n}}\right) = \widetilde{\mathcal{O}}(n^{-\xi/2}) \to 0.$$

As such, all terms go to zero with $n$ (excepting $\mathcal{K}_{\mathrm{bin}}(p_y, \widehat{\phi}_n) \leq \epsilon/2$, which is fine), and there exists $N_0$ so that for all $n > N_0$, all conditions of the bound are met, and with the exclusion of a failure probability of $\delta^{(n)}$, the bound implies $\mathcal{R}(\widehat{W}_n) < \overline{\mathcal{R}} + \epsilon$. Thus $n \geq N_0$ implies $\Pr[E_n] \leq \delta^{(n)} = 1/n^2$, and

$$\sum_{n \geq 1} \Pr[E_n] \leq \sum_{n \leq N_0} 1 + \sum_{n > N_0} \frac{1}{n^2} \leq N_0 + \frac{\pi^2}{6} < \infty,$$

which establishes eq. (10) and completes the proof. $\qquad\square$

## C   Proof of Proposition 1.2

Proposition 1.2 is a consequence of the following more refined statement, which also suggests the method of proof, and is consistent with Figure 2.

**Lemma C.1.** *Suppose marginal distribution $\mu_x$ is continuous and compactly supported on $[0,1]$, $p_y$ is continuous, and that either $\mu_x(p_y^{-1}((0,1/2))) > 0$ or $\mu_x(p_y^{-1}((1/2,1)) > 0$, meaning $p_y$ is outside $\{0, 1/2, 1\}$ on a set which has positive measure according to $\mu_x$.*

*Then there exists a constant $c \in (0, 1/4)$ (depending only on $\mu_x$ and $p_y$) so that with probability at least $1 - 7\delta$ over the draw of $((x_i, y_i))_{i=1}^n$ with $n \geq \ln(1/\delta)/c$, there exists an interval $I \subseteq [0,1]$, and a subset of pairs of indices indices $S \subseteq [m]^2$ satisfying the following properties.*

1. *Either $p_y \in [c, 1/2 - c]$ everywhere on $I$, or $p_y \in [1/2 + c, 1 - c]$ everywhere on $I$; henceforth let $\hat{y} := \mathrm{sgn}(p_y - 1/2)$ designate the correct (Bayes) prediction over $I$.*

2. *If $(i, k) \in S$, then $x_i < x_k = \min\{x_s : x_s \geq x_i\}$, meaning $x_k$ is the first point to the right of $x_i$, and moreover the corresponding labels $y_i = y_k = -\hat{y}$ agree with each other but are incorrect.*

3. *For any local interpolation rule $f \in \mathcal{F}_n$ (cf. Proposition 1.2),*
$$\mathcal{R}_{\mathrm{z}}(f) \geq \overline{\mathcal{R}}_{\mathrm{z}} + c.$$

*Proof of Lemma C.1 (and simultaneously Proposition 1.2).* Consider any point $x$ where $p_y(x) \notin \{0, 1/2, 1\}$ and $\mu_x > 0$; such a point must exist by the assumptions. Define $\hat{y} := \mathrm{sgn}(p_y(x) - 1/2)$ and $c_1 := \min\{p_y(x)/2, |p_y(x) - 1/2|/2, (1 - p_y(x))/2\}$, where $c_1 \in (0, 1/4)$ by construction. Since $p_y$ and $\mu_x$ are continuous, then there must exist some (potentially tiny) closed interval $I$ containing $x$ so that $\mathrm{sgn}(p_y(x) - 1/2) = \hat{y}$, and for any $x' \in I$, both $\mu_x(x') > 0$ and $p_{x'} \in (c_1, 1/2 - c_1) \cup (1/2 + c_1, 1 - c_1)$.

To simplify the rest of the proof, suppose $\hat{y} = -1$; the other case is symmetric, but as in the preceding paragraph, handling both cases simultaneously adds significant notational overhead.

Let $S$ denote all adjacent pairs of points in $I$ where $(x_i, x_k) \in S$ means $x_i < x_k = \min\{x_s : x_s > x_i\}$ and $y_i = y_k = -\hat{y}$. With this choice, all that remains to be shown is the third item, the lower

bound on the risk. To show this, it suffices to show that a constant fraction of $\mu_x$'s probability mass is contained between these pairs, meaning

$$\mu_x\left(\cup_{(i,k)\in S}\mu([x_i, x_k])\right) \geq c_2 > 0,$$

where crucially $c_2$ is independent of $n$. To see that this suffices to establish the third property, suppose that $f : \mathbb{R} \to \mathbb{R}$ satisfies the required condition, meaning $f(x)\hat{y} < 0$ for $x \in \cup_{(i,k)\in S}\mu([x_i, x_k])$; then by a standard calculation against the Bayes risk [Devroye et al., 1996],

$$\mathcal{R}_z(f) - \overline{\mathcal{R}}_z = \int |1 - 2p_y(x)|\mathbb{1}\left[\operatorname{sgn}(f) \neq \operatorname{sgn}(p_y(x) - 1/2)\right] d\mu_x(x)$$

$$\geq \int |1 - 2p_y(x)|\mathbb{1}\left[x \in \cup_{(i,k)\in S}[x_i, x_k]\right] d\mu_x(x)$$

$$\geq 2c_1\mu_x\left(\cup_{(i,k)\in S}[x_i, x_k]\right)$$

$$= 2c_1 c_2,$$

and the final statement and all properties are satisfied if we pick $c \in \left(0, \min\{c_1, c_2, 2c_1 c_2\}\right]$.

As such, it remains to provide a lower bound on $c_2$ which is independent of $n$, which will follow a series of simplifications as follows.

The first step is to lower bound the cardinality of $S$. The expected number of points in $I$ is $n\mu_x(I)$, and if $n \geq 32\ln(1/\delta)/\mu_x(I)$, then by a multiplicative Chernoff bound [Blum et al., 2020, Theorem 12.6], with probability at least $1 - 3\delta$,

$$\left|\{i \in [m] : x_i \in I\}\right| \geq \frac{n\mu_x(I)}{2}.$$

and thus the number of consecutive pairs in $I$ is at least $n\mu_x(I)/2 - 1 \geq n\mu_x(I)/4$.

Since these pairs may share endpoints, consider the set of at least $n\mu_x(I)/8$ pairs that share no points. Since the draw of $y$ is independent of $x$, for each of these consecutive pairs, the probability that both labels are wrong is at least $(1 - c_1)^2$ (and is independent of other pairs), meaning the expected number of such points is at least $n\mu_x(I)(1 - c_1)^2/8$; as such, if $n \geq 256\ln(1/\delta)/(\mu_x(I)(1 - c_1)^2)$, by another multiplicative Chernoff bound, with probability at least $1 - 3\delta$, the number of pairs with agreeing but incorrect labels is at least $n\mu_x(I)(1 - c_1)^2/16$. Let $S_0$ denote this set of pairs; by construction, its cardinality also lower bounds that of $S$.

It remains to show that the union of the convex hulls of these pairs of points has a significant fraction of total probability mass.

For any sample $(x_1, \ldots, x_n)$, let $(x_{(1)}, \ldots, x_{(n)})$ be the sample in sorted order, meaning $x_{(1)} < x_{(2)} < \cdots < x_{(n)}$ (strict inequalities almost surely since $\mu_x$ is continuous). Define a distance $\Delta$ and function $F$ of the sample as

$$\Delta := \frac{\mu_x(I)(1 - c_1)^2}{256n},$$

$$F(x_1, \ldots, x_n) := \left|\left\{i \in [m - 1] : \mu([x_{(i)}, x_{(i+1)}]) < \Delta\right\}\right|;$$

that is to say, $F$ measures the number of consecutive pairs whose convex hulls have probability mass strictly less than $\Delta$. As will be established momentarily, $F$ satisfies the bounded differences property with a constant 2, meaning for any two samples $(x_1, \ldots, x_n)$ and $(x_1', \ldots, x_n')$ that differ only in a single example $x_i \neq x_i'$,

$$\left|F(x_1, \ldots, x_n) - F(x_1', \ldots, x_n')\right| \leq 2.$$

To argue this, suppose the disagreeing example $x_i$ occupies position $j$ after sorting, meaning $x_i = x_{(j)}$, and consider adjusting one sample to the other by renaming this point to $x_i'$, removing it from its current location, and moving it to its final location.

- First we remove $x_i'$ from the interval $(x_{(j-1)}, x_{(j+1)})$. If neither $(x_{(j-1)}, x_i')$ nor $(x_i', x_{(j+1)})$ counts towards $F$, then neither will $(x_{(j-1)}, x_{(j+1)})$, so $F$ remains unchanged. If exactly

one of $(x_{(j-1)}, x_i')$ and $(x_i', x_{(j+1)})$ counts towards $F$, then $(x_{(j-1)}, x_{(j+1)})$ does not count towards $F$, so $F$ decreases by 1. If both $(x_{(j-1)}, x_i')$ and $(x_i', x_{(j+1)})$ counts towards $F$, then $(x_{(j-1)}, x_{(j+1)})$ may or may not count towards $F$, so $F$ decreases by 1 or 2. So this operation changes $F$ by any of $\{-2, -1, 0\}$.

- Then we insert $x_i'$ into a new interval. The range of possible changes to $F$ is the exact opposite as removing it from an interval, so this leads to a change by any of $\{+2, +1, 0\}$; together the difference in $F$ is within $[-2, +2]$.

As such, by McDiarmid's inequality, with probability at least $1 - \delta$,

$$F(x_1, \ldots, x_n) \leq \mathbb{E}F(x_1, \ldots, x_n) + \sqrt{2n \ln(1/\delta)}.$$

Upper bounding $\mathbb{E}F(x_1, \ldots, x_n)$ can now be performed in a coarse way as follows. Partition the support of $\mu_x$, $[0, 1]$, into two systems of intervals, $\mathcal{I}$ and $\mathcal{J}$, as follows. $\mathcal{I}$ simply contains the $\lceil 1/(2\Delta) \rceil$ consecutive intervals of mass $2\Delta$ (except for the last, which may have less mass); meanwhile, $\mathcal{J}$ contains a first initial interval of mass $\Delta$, and then intervals of mass $2\Delta$ until a final interval of mass at most $2\Delta$. Due to this staggered behavior, if some pair $(x_{(i)}, x_{(i+1)})$ has $\mu_x((x_{(i)}, x_{(i+1)})) < \Delta$, then the pair must appear in a single interval in either $\mathcal{I}$ or $\mathcal{J}$ (the staggering avoids boundary issues). Now consider the creation of the full data sample by sampling the data points one by one, and the resulting effect on these bins; the goal is to upper bound the number of times a point is inserted into an occupied bin, as this upper bounds the number of consecutive pairs of points within some bin, which in turn upper bounds $F$. After inserting the $i$th point (twice), let $A_i$ denote the number of occupied bins, and $B_i$ the number of times a point was inserted into an occupied bin; necessarily, $A_i = 2i - B_i$ (the factor two coming from simultaneous throws to $\mathcal{I}$ and $\mathcal{J}$). The probability of landing in an occupied bin (and thus increasing $B_i$) is at most $A_i(2\Delta) = (2i - B_i)(2\Delta)$. By linearity of expectation,

$$\mathbb{E}F \leq \mathbb{E}B_n \leq \sum_{i=1}^{n-1} 2\mathbb{E}\mathbb{1}\left[x_{i+1} \text{ lands in an occupied bin}\right]$$

$$\leq 4\Delta \sum_{i=1}^{n-1}(2i - \mathbb{E}B_i) \leq 4\Delta(n-1)n \leq \frac{n\mu_x(I)(1-c_1)^2}{64}.$$

Together, supposing that $n \geq 8192 \ln(1/\delta)/(\mu_x(I)^2(1-c_1)^4)$, it follows that with probability at least $1 - \delta$,

$$F(x_1, \ldots, x_n) \leq \frac{n\mu_x(I)(1-c_1)^2}{64} + \sqrt{2n \ln(1/\delta)} \leq \frac{n\mu_x(I)(1-c_1)^2}{32}.$$

To finish the proof, since the preceding quantity is less than half the cardinality of $S_0$, we are guaranteed that at least half the pairs in $S_0$ have $\mu_x((x_i, x_k)) \geq \Delta$; letting $S_1$ denote this half, then

$$\mu_x(\cup_{i,k \in S}[x_i, x_k]) \geq \sum_{(i,k) \in S_1} \mu_x([x_i, x_k])$$

$$\geq |S_1|\Delta \geq \frac{n\mu_x(I)(1-c_1)^2}{32} \cdot \Delta \geq \frac{\mu_x(I)^2(1-c_1)^4}{8192} =: c_3.$$

It only remains to determine the final value of the constant $c$. By the preceding calculation and the comments near the start of the proof establishing that $c \in (0, \min\{c_1, c_2, 2c_1c_2\}]$ suffices, the quantity $c_3$ here is indeed a lower bound on $c_2$, and thus, defining $c_4 := \min\{c_1, c_3, 2c_1c_3\}$, it suffices to require $c \in (0, c_4]$. On the other hand, inspecting all the necessary lower bounds on $n$ throughout the proof, the maximum across all of them is that we need $n \geq \ln(1/\delta)/c_3$. As such, all properties are satisfied if we take $c := c_4 > 0$ as our final constant, which depends only on $\mu_x$ and $p_y$ (but not on $n$) as promised. $\square$