# OpenReview forum: "Early-stopped neural networks are consistent"
_NeurIPS.cc/2021/Conference — NeurIPS 2021 Spotlight_

### Official Review · Reviewer_9b2J · 2021-07-14

**Rating:** 7
**Confidence:** 1

**Summary:**

This paper studies the training of single layer neural networks trained with gradient descent. The authors provide an upper bound on the logistic loss function using a reference infinite-width random feature model (which does not depend on the sampling of the training set). They also show that if a classifier in the 1-dimensional setting is in the interpolating regime, then the classification error is (depending on the number of consecutive wrong labeled samples) large.

The proof is lengthy and quite involved and in appendix. On the one hand, a sketch of each result is provided in the main paper: it sheds light on the novel ingredients and technicalities used in the proof, which is useful (and explains for instance why the paper is restricted to logistic losses). On the other hand, the paper could make a better job to explain the gist behind the main result in (Theorem 1, which is half a page long just to be stated) in Section 1, by complementing the interesting remark 1.1 with a more practical discussion on how the parameters $m,\rho, R, \delta, \epsilon_{gd}$ are to be chosen as a function of $n, d$ to bring the three terms of the bound close to zero (simultaneously), and how the reference model $\overline{U}_\infty$ is obtained from the data. An interesting feature of this bound is indeed that it decomposes in three errors (on the model, optimization convergence, and generalization).

While the results are definitely worth publishing, I wonder if a journal in ML theory (where enough time is given to review the proofs in appendix) might not be a more appropriate venue than a conference for such a purely theoretical paper.

**Limitations And Societal Impact:**

Not relevant, the paper is purely mathematical on a theoretical model.

**Main Review:**

The proof of Theorem 1.1, as stated in line 107, lets the training iterations increase. Yet the classifier chosen in Theorem 1.1 is the one with the lowest empirical (training) loss ($R(W_{\leq t})$). The authors refer to this as the early stopped model, but the model is more the best model obtained on the training data up to time $t$. However, in practice an early stopped model does not have the lowest training loss: a low training loss is avoided by early stopping, which should not interpolate all the training samples and therefore the noise that could be present in the training set. Could you make the connection between the model $W_{\leq t}$ and early stopping?

Minor nit: In the equation between the lines 41 and 42: $l(r)$ should be $l(z)$.



**Time Spent Reviewing:**

4

---

> ### Author Response · Authors · 2021-08-10
> **Thank you for your review.**
>
>
> We thank the reviewer for their comments and time, and agree with all points we
> do not explicitly mention.  Here are our main responses.
>
> 1. **Presentation of Theorem 1.1.** We agree that it is hard to interpret the
>    statement, and even to ensure that terms really can all be mutually
>    controlled.  We have considered making Theorem 1.1 an abstract lemma
>    and leaving it in the appendix, and instead having two separate theorems
>    (corollaries) in the body:
>    a "worst case" theorem where we fix many terms like $\rho$ and explicitly
>    compute $\tau_0$, $\tau_1$, and get a bound which clearly goes to $0$ with
>    $m$ and $n$ for arbitrary data, and a separate "nice data" theorem where
>    instead we assume a nice condition on $\overline{U}_\infty$ and obtain
>    a clean, interpretable faster rate.
>    We could also move much of the words/presentation of Theorem 1.1 into the
>    introduction and related discussion.
>    We are eager to hear more feedback from the reviewer on this approach.
>
> 1. **Journal or conference?** As the main prior and relevant work appears in conferences,
>   we feel NeurIPS is the ideal venue.  We will work to improve the presentation
>   as the reviewer suggests.
>
> 1. **Issues with the relationship between early-stopping and interpolation.**
>   We agree that our our discussion here was inadequate and will expand the body;
>   briefly, here are a few comments which reflect our understanding, and we
>   are eager to hear more comments from the reviewer!
>
>     First, two technical points:
>
>     - Early stopping refers to the selection of $t$, namely the maximum iteration
>       considered by the argmin within $W_{\leq t}$.  As in, while $W_{\leq t}$ is chosen
>       via training error, we only consider iterations up to some $t$, and we can not
>       let this $t$ be too large in general.
>
>     - Secondly, as we discuss in the general comment above, we had an oversight in
>       Theorem 1.1, we need to replace $R$ in a few places in the statement
>       with $R + \sqrt{ t \cdot \mathcal{R}(\overline{U}_\infty)}$.  In particular,
>       this means the generalization part of the full bound has a
>       term $\sqrt{ t \cdot \mathcal{R}(\overline{U}_\infty) / n}$, which explicitly
>       requires early stopping whenever $\mathcal{R}(\overline{U}_\infty)$ is not close to
>       zero in order to ensure $t/n = o(1)$.  As is discussed there, after this fix we still have our "data simplicity" story, and also we can still control all terms and establish our consistency corollary.
>
>     With these technical points out of the way, we summarize the early-stopping situation
>     as follows.  In the general case, we will need $t=o(n)$, and may fail to interpolate;
>     this is certainly necessary to control all terms in our upper bound (Theorem 1.1), but
>     we don't fully prove it is necessary (our Proposition 1.2 is specialized, as many reviewers
>     point out).
>
>     We are eager to discuss further, and look forward to any other comments by the reviewer!

---

> > ### Comment · Reviewer_9b2J · 2021-08-18
> > **Updated feedback**
> >
> > Thanks for the clarification under "Experiments, magic quantities, and non-vacuous bounds": these would be actually very helpful. It's good to see strong theory papers in the NeurIPS program, but as Theorem 1.1. comes with a quite high descriptive complexity, I find it even more important to include these clarifications in the main part for a conference paper than a journal paper.
> >
> > It's good also to fix the oversight in Theorem 1.1 - and here I have to trust that the proofs in appendix will be fine for the final version. I'm afraid that the lack of time for reviewers to carefully check proofs in appendix leaves this task with the authors... which is why a journal in ML theory has better archival value for me for such a contribution, than a conference.

---

> > > ### Author Response · Authors · 2021-08-20
> > > **.**
> > >
> > > Thank you for going through our response, and for your support of our work. We will gladly answer any further questions.

---

### Official Review · Reviewer_oWmh · 2021-07-15

**Rating:** 6
**Confidence:** 4

**Summary:**

This paper gives a theoretical analysis of the neural networks under the logistic loss on binary classification task. It considers the setting of training a shallow two-layer neural network with ReLU activation, and proves a core theorem that gives a high probability bound on the difference between population risk and Bayes risk. The obtained results imply consistency, since all parameters can be made arbitrarily small. It also proves a proposition related to 1d datasets, where if the neural network interpolates the training data perfectly and is not too wild between training points, then it is guaranteed to perform badly on noisy test data.

**Limitations And Societal Impact:**

Yes, it discusses limitations. It does not discuss societal impact, however, since it's a pure theory paper, it does not have a direct impact.

**Main Review:**

Strength:

1. This paper gives a very strong theoretical result regarding the impact of early-stopping on training a shallow neural network. Specifically, it associates a multiplicative factor on the term R(\overline U_\infty), suppose this term is smaller than eps_gd then \tau_0 can be ignored and the KL divergence term is also smaller than eps_gd. This recovers a result for data separable setting [Ji and Telgarsky, ICLR’2020] but generalizes it into the non-separable setting.

2. The technical lemma regarding comparing a neural network to a local linear approximation has been improved, rather than an additive error term as in prior works, it yields a multiplicative error term. This is the key to ensure this paper to get a similar result as in data separable setting.

3. To prove the linearization approximation, it bounds an expression \langle \nabla f(x;W)-\nabla f(x;W_0), W \rangle for all \|x\| \leq 1. This is in contrast to the standard practice where the quantity f(x;V)-f(x;W)- \langle \nabla f(x;W), V-W \rangle for a fixed data point x and use a union bound. The technique employed in this paper is novel and from a different perspective compared to the convergence analysis of gradient descent on over-parametrized neural networks [Allen-Zhu, Li and Song, ICML’2019]. This co-VC dimension inspired technique has potentially to be used in other analysis, e.g., the convergence analysis of NTK and other variants.

Weakness:

1. The paper layout is a bit unusual and hard to follow. It spends a very small amount of paragraphs to set up the problem tentatively, then discuss the major contributions. These contributions are discussed in more details later with some of the notations left undefined until a later section. Section 2 and 3 are dedicated to present the essentials of the proof. It might be better to first introduce the problem background, its formal setup and some of the related literature on this problem in more detail. Then present a preliminary version of the main result. More details of the result can be laid out after introducing the notations.

2. It might be better to summarize and compare the results obtained in this paper with various previous works under different settings. For example, this paper obtains a similar popular risk loss result for data non-separable setting as some early works addressing data separable setting. Also, what’s the difference between the results obtained using the co-VC dimension technique compared with the standard analysis which makes use of the fact that weights are not far away from the initialization?

3. The early-stopping discussion in this paper is very preliminary. It is suggested on line 132 that this is due to technical reasons in the proof of Theorem 1.1, however, I don’t see an explicit mention or discussion related to early-stopping for this Theorem. A negative 1d result is also discussed as in Proposition 1.2. Due to the limited paragraphs involving this topic, it is hard to understand the importance and impact of early-stopping.

If authors can address the above 2 and 3, and provide further clarification on the subject, I can raise the score.

Typos: On line 144, the definition of {\cal F}_n should be f: \R\rightarrow \R since the proposition considers 1d data.

Minor:

Some missing NTK citations:

Quadratic suffices for over-parametrization via matrix Chernoff bound.

On the Convergence Rate of Training Recurrent Neural Networks.

Training (over-parameterized) neural network in near linear time.


**Time Spent Reviewing:**

10

---

> ### Author Response · Authors · 2021-08-10
> **Thank you for your review.**
>
> We thank the reviewer for their comments and time, and agree with all points we
> do not explicitly mention.  Here are our main responses.
>
> 1. **Unusual layout.** We agree that the paper can be more direct presenting
>    its key statements, contributions, and comparison to prior work.  We tried
>    for a "relaxed" presentation and agree it was sometimes "tentative", as the
>    reviewer says.  One aspect we wish to improve, which the reviewer did not
>    mention, is better highlighting the new elements in the proof summary, e.g.,
>    the explicit new linearization technique and its key proof aspects.  (We
>    highlighted some of these contributions in our response to reviewer afNw.)
>
> 1. **Comparison to works using weights near initialization.**
>    We are not sure we follow this question, since our proofs also stay near
>    initialization, but here is our best understanding and we urge the reviewer
>    to follow up.
>
>    Firstly, we also stay near initialization.  However, we may even stop much
>    earlier.  As a concrete example, suppose the data has some regularity (e.g.,
>    we never sample the same data point twice with opposite labels), and
>    consider a very large width.
>    Then we can consider a standard NTK analysis, e.g., by Allen-Zhu/Li/Song,
>    and achieve training error zero.  The test error after such an analysis will
>    be unclear, however.  Meanwhile our proof technique will suggest stopping earlier,
>    possibly with non-zero training error, and achieve a good test error.
>
>    This point is quite delicate and we are eager to discuss it further!
>
> 1. **Confusing discussion of early-stopping.**
>   We agree we need to expand our discussion here; we even considered including a figure with the
>   gradient descent path and different regimes (e.g., including feature learning), and are open
>   to further comments.  In more detail, our thoughts are as follows.
>
>     Firstly, we feel some of the confusion is due to a term we neglected to include in Theorem 1.1.
>     We detail this term in the general remark above, but as a brief summary, the correction
>     is to replace $R$ with $R + \sqrt{ t \cdot \mathcal{R}(\overline{U}_\infty)}$ in a few places.
>     This term can be interpreted as "either a good test error is possible, or our optimization radius
>     can blow up as $\sqrt{t}$".  (We detail the fix to the proof above, but it is very easy,
>     just some brief algebra inside one existing lemma and no new machinery.)
>     Once this term is corrected, then Theorem 1.1 has an explicit
>     $\sqrt{ t \cdot \mathcal{R}(\overline{U}_\infty) / n}$ term, which means we must require $t = o(n)$
>     whenever $\mathcal{R}(\overline{U}_\infty)$ is not small.
>     As such, the bound still achieves our goals: it still reflects data simplicity, and still all terms can be controlled to prove consistency as a corollary (more details are in the general comment).
>     Does this correction clarify the picture?
>
>     Overall, we summarize our understanding of early stopping as follows.
>     Our upper bound proof technique, with the correction, requires early stoppping;
>     it also requires weights to stay near initialization, but in fact we might
>     stop much earlier than other analyses near initialization, and not achieve
>     zero training error, but on the other hand we achieve good test error.
>     Our lower bound technique, meanwhile, is far from establishing the necessity
>     of early stopping; we will discuss this gap, as it is a valuable direction
>     for future work.
>
>     We will gladly include these points in our revisions, and are eager for further comments
>     from the reviewer!
>
>
> 1. **Missing citations.**
>    We thank the reviewer very much for mentioning these
>    very interesting works. We'll revise the related work to discuss their
>    relationship to our paper.
>    "Quadratic suffices [...]" in particular is very relevant to our interest in
>    handling the smallest possible widths.
>    Thank you again!

---

### Official Review · Reviewer_sDSa · 2021-07-16

**Rating:** 8
**Confidence:** 4

**Summary:**

This work shows that a certain neural network learning procedure
(a single trainable layer, trained via gradient descent)
is consistent, meaning it asymptotically converges to the Bayes optimal classifier.
Although certain algorithms on neural networks were known to be consistent before, this result was not know for *gradient descent*
from random initialization.
This paper thus contributes to our fundamental understanding
of the power of gradient methods on neural networks.

**Limitations And Societal Impact:**

Suggestions on limitations are in the "weakness" section above.
Societal impacts N/A.

**Main Review:**

## Recommendation

I recommend acceptance.
The result is clean, the paper is very well-written,
and the conclusion is significant.

As a disclaimer, I did not get a chance to check all of the proofs.
However, the overall proof strategy appears sound.

## Strengths

- The writing is clear, and outlines the proof strategy.
- The result is, as far as I know, not known before.
(Although the tools involved are used in many related papers, as is acknowledged).
- The result is "good to know": gradient methods are often used in practice, and having such consistency results can be a first step towards justifying them.
- The theoretical requirement of early-stopping may be related to certain relevant practical phenomenon, and could shed light on differences between interpolating and non-interpolating classifiers.

## Weaknesses

- As with all consistency results, it is not clear how much this result says about why neural networks
are actually successful in practice.
(For example, k-nearest-neighbors are also asymptotically consistent...). The quantitative bounds on generalization may very well diverge exponentially fast, depending on the problem setting.
In general, "consistency" is a fairly weak property, but it is still good to establish.

- The results on "adaptivity to data simplicity" does not seem like a major contribution, since results of this form were already known in the past (as the paper indeed cites), for various notions of "simplicity." And it is not clear how this result contributes beyond what is already known in the area.

### Relevant Citations

The following empirical paper shows that, complementary to your results, interpolating networks are *not* consistent on noisy distributions:
https://arxiv.org/abs/2009.08092

This in particular addresses your stated open question of
"Do neural networks fail on noisy data if care is not taken with temperature and early stopping?"

Theorem 1 in the above paper may also be seen as a certain analogue of your Proposition 1.2, for the case of nearest-neighbor classifiers.
In general, the fact that 1-nearest-neighbors is not asymptotically consistent is related to your claims about local interpolation rules.


----
Edit after rebuttal: score increased from 7 to 8.

**Time Spent Reviewing:**

3

---

> ### Author Response · Authors · 2021-08-10
> **Thank you for your review.**
>
>
> We thank the reviewer for their comments and time, and agree with all points we
> do not explicitly mention.  Here are our main responses.
>
> 1. **Unclear if relevant to practical success of deep networks.**
>    Overall we agree with this point, and will expand our "open problem" section to further
>    close this gap, and include comments in the body.
>    Regarding $k$-nn, we mention one interesting point, that convergence rates
>    for $k$-nn are non-parametric (to our knowledge), whereas our rate here is
>    a parametric $1/\sqrt{n}$ rate, as with boosting methods; we feel this deserves
>    further study.
>    Relatedly, while it is true that our work can be viewed as a sanity check,
>    we had no idea if it was possible before completing all proofs.
>    As an example, our proofs can handle arbitrarily large widths (Theorem 1.1
>    only places lower bounds on width), whereas most generalization bounds blow
>    up with large width, something we sidestep in this general setting
>    via our "delinearization" technique as detailed to reviewer afNw.
>
> 1. **Data simplicity not a major contribution.**
>   While our characterization can be improved, we do not know of a general
>   analysis that handles data simplicity.  E.g., while the infinite-width
>   simplicity notion appeared in the prior work of Ji & Telgarsky, they require
>   margins, and their proof technique can not handle data with negative
>   margins as we require here.
>   A similar data simplicity notion was considered in the follow-up
>   work we cite by Chen et al., which moreover can handle some small amount of
>   margin violations, however firstly they do not use infinite-width
>   reference networks, and secondly their test error has a multiplicative
>   constant which means they can not achieve optimal test error. As detailed to
>   reviewer afNW, our simplicity notion is sufficient to imply some non-vacuous
>   bounds, and thus we feel it is promising.
>
> 1. **Citing Nakkiran & Bansal.**  We agree this is very relevant, and apologize
>    for the omission!  Concretely, the experiments with noise can be included
>    both in the related work where we mention works of Bai et al., which
>    theoretically analyze noise, and perhaps we can also incorporate the
>    citation earlier in the body for motivation.  We can also mention the
>    relationship with Proposition 1.2, though it is a bit delicate, as the
>    settings and proof techniques differ.

---

> > ### Comment · Reviewer_sDSa · 2021-08-14
> > **reply**
> >
> > Thank you for your response. Upon reflection, I have increased my score from 7 to 8.

---

> > > ### Author Response · Authors · 2021-08-20
> > > **.**
> > >
> > > Thank you for going through our response, and for your support of our work.  We will gladly answer any further questions.

---

### Official Review · Reviewer_PKyu · 2021-07-23

**Rating:** 8
**Confidence:** 2

**Summary:**

The paper studies 1-hidden layer ReLU networks trained with SGD and logistic loss on arbitrary classification tasks, showing optimality given that early-stopping is adopted. An interesting complexity measure for the underlying mapping is shown to play a key role in how the generalization error is affected by relevant factors such as number of iterations and network neurons. Insights on interpolation capacity of networks are given along with new and exciting questions.

**Limitations And Societal Impact:**

Yes.

**Main Review:**

The paper is very well-written, clear, and the discussion on both implications and limitations of the analysis is impeccable.

I see the main Theorem as a strong contribution, showing optimality of shallow ReLU networks w.r.t. the population loss under very mild assumptions. The need for early-stopping for the main result is thought-provoking, and the provided discussion on its relation to the interpolation regime and NTK (where both have gained a lot of attention recently) draw exciting connections and open new questions for future research.

The complexity measure presented here is also exciting, as it seems to interact in a very desirable way with the relevant model & training factors and is also seems intuitive and is strongly connected to the NTK. A key question is how much the behavior changes once actual learning is taking place, but I can see this analysis requiring considerable additional technical effort and it being appropriate to leave it completely to future work.

Although I haven't checked the proofs in detail, it seems that there is considerable technical contribution as well.

I believe this is a strong submission, with a solid contribution which is presented extremely well and raises interesting questions, therefore I recommend acceptance. However, I would like to note that I am not very familiar with the topic and might have missed possible limitations / concerns.

**Time Spent Reviewing:**

3

---

> ### Author Response · Authors · 2021-08-10
> **Thank you for your review.**
>
> We thank the reviewer for their comments, time, and for their support of our work.
>
> If the reviewer has any questions during the remaining review period, we would
> be eager to discuss them.
>
> Regarding the point of how training proceeds once the feature learning regime
> is entered, of course we would love to have a general result there.  We discuss
> this twice in the open problems: one interesting though technical point
> is that we feel the temperature does not need to be carefully chosen once
> feature learning occurs.  Another funny point is that our "early-stopping" is
> in fact "extreme early stopping": one must early stop to stay within the NTK
> regime, but in fact, with a large width, one can guarantee 0 training error,
> and for noisy cases our analysis will stop even earlier than the
> early-stopping that ensures staying within the NTK, since we do not want
> training error 0!  As per other reviews, we plan to expand our discussion
> of early-stopping and clarify these points.
>
> Thank you again for your support!

---

### Official Review · Reviewer_afNw · 2021-08-02

**Rating:** 6
**Confidence:** 3

**Summary:**

The paper studies the convergence properties of the gradient descent (GD) algorithm for training two layers neural networks. The convergence is established wrt the population loss and under essentially no constraint on the underlying data distribution. Moreover, the simplicity of a given data distribution is determined by measuring the complexity of infinite-width random feature models which approximate the underlying distribution. It is shown then that less complex is the data the less samples, network nodes and iterations are needed.


**Limitations And Societal Impact:**

Yes, the authors have adequately addressed the limitations and potential negative societal impact of their work.


**Main Review:**

One of the current great challenges in deep learning is to understand the ability of GD to converge to models which exhibit good generalization capabilities--despite the nonconvexity of the training loss. Thus, the strong convergence results established in this work are rather interesting. In addition, tying the convergence properties of GD to the structure of the underlying data distribution seems to be a step in the right direction of abandoning\refining the distribution-free paradigm presently governing the machine learning literature, and I find it very it encouraging.

To me, the two main issues are: 1. large parts of the analysis seem rather incremental when compared to previous works. 2. The positioning of early-stopping, the main selling point of the paper, are not entirely clear: the negative result (possible failures of DL learning in the interpolation regime) addresses a nearly-negligible part of the regime to which the positive result (convergence rates of GD) applies. Thus, I believe that the analysis of early-stopping in the paper cannot be regarded as conclusive.

Assorted comments:
* I feel that, due to their importance in the paper, the very notion of early-stopping and interpolation regime (the phenomenon of double descent, in particular) should be introduced and discussed more explicitly.

* L41, the predictor function f is not explicitly introduced.

* Perhaps discuss the implications of the modeling choice of drawing the weights of the second layer at random more thoroughly. In particular, can you characterize a large deterministic set of 2nd-layer weights to which similar convergence results apply?

* Theorem 1.1, use of n instead of m (above L96).

* Theorem 1.1, is there a practical way of estimating the quantities involved in the upper and lower bound presented in the theorem for a number of interesting settings to see if they behave nicely? In the nearby realm of sample complexity, bounds are, by and large, vacuous when instantiated to practical settings. See, e.g., https://arxiv.org/abs/1703.11008 (and references therein).

* L151 "recent works whose width"

* L284 :
-- What other models can be used to measure simplicity instead of the infinite-width random features model?
-- What are the upsides and downsides of the current choice?
-- Quantitively, how 'simple' are MNIST, CIFAR and other conventional datasets in ML when measured using the method proposed in the paper?

**Time Spent Reviewing:**

3.5

---

> ### Author Response · Authors · 2021-08-10
> **Thank you for your review.**
>
>
> We thank the reviewer for their comments and time, and agree with all points we
> do not explicitly mention.  Here are our main responses.
>
> 1. **Large parts of the analysis seem rather incremental.**  While we did rely on a number of
>   existing tools, we have a number of new components which are central to our story.  We will
>   adjust our body, in particular the proof sketch, to better highlight the new contributions, but
>   here is a brief summary.
>
>     - In section 2.1 we mentioned a "co-VC dimension" technique.  Concretely, this is item 2 of Lemma A.5,
>       but most of the heavy lifting is in Lemma A.4.  Lemma A.4 is nontrivial and not based on any argument
>       we know in the literature, and is the main technical advance over prior work.  This Lemma
>       in turn powers many of other key steps in the proof, and we do not know how to perform these steps
>       without it:
>
>         1. We mentioned that generalization proceeds via linearization and
>            "de-linearization".  This second step is over the distribution, not just
>            a finite sample, and makes crucial use of de-linearization over the
>            entire sphere. We only know how to do this using the aforementioned
>            Lemma A.5, and not anything from prior work.
>
>         1. When we sample the infinite width model $\overline{U}_\infty$ to produce a good finite-width
>           reference network, we again need to control the error over the whole sphere, and use Lemma A.5.
>           Prior work provides no alternative; e.g., while the work of Ji & Telgarsky also samples infinite-width
>           networks, there they have a margin, and it suffices to do an elementary sampling analysis
>           which only considers a finite training set.
>
>     - As minor technical contributions which are not direct from prior work: (a) the multiplicative property
>       of the logistic loss, as in Lemma 2.1, and in particular the consequence that the optimization analysis can be
>       carried through invoking only these multiplicative errors; (b) the idea to linearize and de-linearize
>       for generalization has not appeared before.
>
> 1. **The analysis of early-stoping [...] cannot be regarded as conclusive.** We
>    agree; e.g., we only showed early-stopping is sufficient, but we only
>    suggested (via Proposition 1.2) that it is necessary, and did not fully
>    establish necessity. We will add this explicitly as an open problem in
>    Section 4, and expand our discussion in the rest of the body.
>
> 1. **Interpolation and double descent.**  We agree, though we remark that the
>    discussion will be delicate, since the main works in that literature
>    optimize the squared loss, and often work with specialized data.  E.g.,
>    double descent usually has its "peak" when $m=n$ (our width $m$ corresponds
>    to the dimension there), whereas here we have situations where we can
>    interpolate with $m = \text{polylog}(n)$.  And for example, with
>    interpolation, as in our Proposition 1.2, often we will early stop far
>    before interpolation occurs.  If the reviewer would like to suggest
>    a concrete investigation, we would be happy to follow it.  E.g., we could
>    take the precise setting of standard double descent (e.g., Gaussian data,
>    and the noise model), and point out where different choices in our theorem
>    (e.g., $m$) fall on the curve.
>
> 1. **Choices for second layer weights.** Basically anything with similar
>    statistics to our choice will be fine; e.g., deterministically alternating
>    $\pm 1$ signs, or using Gaussians.
>
> 1. **Estimating magic quantities, in particular for mnist and cifar.**
>   The key magic object in our eyes (please correct us if there was another quantity) is
>   $\overline{U}_\infty$, its norm $R$, and its population risk $\mathcal{R}(\overline{U}_\infty)$.
>   This quantity can in fact be estimated, as we detail in our general response above, but here
>   we will provide a brief summary.  We can train a *finite-width* NTK of a similar form to
>   $\overline{U}_\infty$, and prove that its norm and test error are reflective of a related
>   infinite-width model, thus estimating the missing quantities.  Two key questions and their
>   resolution is as follows.
>
>     - **Does this match our story, at least on mnist or cifar?**  We ran the preceding experiment
>       (as detailed above) on mnist 1 vs 5 and 3 vs 5, and found that the complexity estimates
>       were much better (lower) for 1 vs 5, an intuitively easier prediction problem.
>       Therefore these experiments are consistent with our story.  As detailed above, we did
>       not try hard to optimize these numbers, and feel they can be made even more compelling.
>
>     - **Are the estimates non-vacuous?** We will discuss this in more detail in a separate point,
>       but briefly, the flexibility of Theorem 1.1 is we can effectively *enforce* non-vacuity
>       as follows.  As in the experiment we conducted, and noting that the key complexity term
>       related to $n$ in Theorem 1.1 is $R/\sqrt{n}$, we can consider models where $R = o(\sqrt{n})$.
>       We were still able to obtain meaningful bounds (i.e., subject to this constraint,
>       we were still able to achieve good test error).
>
>     As a brief remark, regarding cifar, since generally it requires a multi-layer and/or
>     convolutional structure for good performance, we did not check how it behaved, but
>     are open to input on this point, especially since mnist is "easy".
>
>     We were very fascinated by these experiments and certainly would like to include
>     a more elaborate version in the paper, and are eager for further comments!
>
> 1. **Comparison to existing generalization bounds.**
>   We thank the reviewer for mentioning the work on "non-vacuous generalization
>   bounds".  Continuing with the previous point, we feel that we did provide non-vacuous
>   bounds, though the situation is a bit delicate, as in the following two points.
>
>     - Firstly, the relationship to existing generalization bounds is delicate.
>       On the one hand, the linearization/de-linearization generalization
>       technique here saves a factor at least $\sqrt{m}$ over the standard
>       Rademacher bounds (say, Bartlett et al. and Golowich et al.). However, the
>       present bound only holds for activation patterns close to the initial one,
>       whereas those others are more general.  To summarize, while on the positive side
>       our generalization bound technique seems to omit extra factors which rendered
>       many previous bounds "non-vacuous", we can not compare them apples-to-oranges
>       since ours only applies to a very restricted subclass of networks which
>       is specialized to our analysis.
>
>     - Secondly, to assert "non-vacuity", we still need to estimate all the magic
>       quantities. As in the experiment above, we chose parameters to ensure we
>       are in the ballpark for a non-vacuous claim: specifically, $R/\sqrt{n} < 1$
>       and $\mathcal{R}(\overline{U}_\infty)\ll 1$.  These experiments were
>       stable as width varied so we expect we can take $m\to\infty$ and wash out
>       those additional factors.
>
>     Overall, these points are very delicate, and we are eager to improve the paper
>     in our revisions and receive further feedback from the reviewer!
>
> 1. **What other models can be used to measure simplicity.**
>   Since our choices are related to the NTK, anything which assesses simplicity there should also work;
>   e.g., we should be able to use Barron norms, and will list this as an open problem.

---

### Author Response · Authors · 2021-08-10
**Detailed comments on two general points; main responses are individual.**

We thank the reviewers for their comments and time.
We respond to each reviewer individually, but here we present in detail
two important points that arose in multiple reviews.


1. **Experiments, magic quantities, and non-vacuous bounds.**  This was mentioned explicitly by reviewers
  afNw and 9b2j, but we feel it is of interest to all reviewers.

    Two key related questions are: how to estimate the key magic quantities in Theorem 1.1
    (meaning
    a choice of $\overline{U}_\infty$ for which the norm $R$ and
    the population risk $\mathcal{R}(\overline{U}_\infty)$ are simultaneously small),
    and relatedly if this leads to a non-vacuous bound for reasonable data.
    Here we detail how these quantities can be estimated on mnist and give a result
    which is consistent with our story and arguably non-vacuous;
    we are certainly eager to discuss both points with the reviewers.

    The summary of how to estimate these quantities from data, and our experimental
    setup, are as follows.

    1. **Calculation from data.**  First we show how any finite-width ntk can be
         used to estimate the above.
         Concretely, suppose weights Gaussian vectors $(z_1,\ldots,z_m)$ are sampled
         and then weights $(v_1,\ldots,v_m)$ are trained in a linear predictor of the form
         $$
          x\mapsto \frac 1 {m} \sum_{j=1}^m a_j (v_j^T x) 1\\!\\!1[z_j^T x \geq 0];
         $$
         here $a_j$ are as in the paper, but the outer factor is $1/m$ not $1/\sqrt{m}$
         to mirror the infinite width model.

         We can use this to construct an infinite width model $U_\infty$ as follows.
         Construct an optimal transport mapping $F:\mathbb{R}^d \to [m]$ from Gaussian random vectors
         over $\mathbb{R}^d$ to the indices of the random vectors $(z_1,\ldots,z_m)$,
         and define $U_\infty(z) := a_j v_j$ where $j = F(z)$.  Then, for any $x$,
         $$
          \int U_\infty(z)^T x 1\\!\\!1[z_{F(z)}^T x\geq 0] d \mathcal{N}(z)
          = \frac 1 m \sum_{j=1}^m a_j v_j^T x 1\\!\\!1 [ z_j^T x\geq 0],
         $$
         which can be used to relate the performance of $U_\infty$ and the finite-width
         network, and to estimate its norm and population risk.
         (The use of an optimal transport mapping keeps these terms small.)

    1. **Example experiment.**
      Relying upon the above calculation,
      we can simply train the finite-width model with gradient descent,
      and then use its norm and test error to reflect
      $R$ and
      $\mathcal{R}(U_{\\infty})$
      for the related infinite-width model $U_{\\infty}$.
      Here is a brief overview of our experimental setup:

        - We ran gradient descent on the above finite-width NTK on two classification
          tasks derived from the mnist data: digits 1 vs 5, which are linearly separable
          and should represent an easy task, and digits 3 vs 5, which are not linearly
          separable and should represent a harder task.

        - We ran a basic gradient descent with the logistic loss and no bells or whistles.
          We ran well past zero training error, but test error continued to decrease.
          Test zero-one loss was smaller than test logistic loss, but we only report the
          latter.

        We ran both until $R$ was around 50; since $n\approx 12,000$ in both cases,
        then the key quantity $R/\sqrt{n}$ still gives us an arguably non-vacuous generalization bound.
        For the "easy" 1 vs 5 task,
        $\\mathcal{R}(U_\\infty)\approx 0.01$ (as estimated by the test set),
        whereas for the harder 3 vs 5 task,
        $\\mathcal{R}(U_\\infty)\approx 0.08$;
        thus the population risk estimate reflects the desired "simplicity".

     We are happy to expand this experiment and include it in the paper, we certainly agree
     such an experiment is important.

     A few more remarks:

     - The above results are using width $64$; we tried other widths briefly and in fact the numbers
       improve.  This also plays nicely with our generalization bound, which was designed to
       not grow with width as with standard bounds.

     - We did not work to bring the numbers down at all, thus much better bounds are possible.

     - It is also interesting to provide lower bounds to fully separate claimed "easy" and "hard"
       instances, but we have not tried this.

     - One can also estimate these quantities for explicit data of various types;
       e.g., the prior work of Ji&Telgarsky analyzed a few convenient data distributions,
       and the end of our Theorem 1.1 gives a worst-case estimate for arbitrary measurable
       target functions.

     We are eager to include these experiments in our revisions, and discuss further with reviewers!

1. **Benefit of early stopping not clear in Theorem 1.1.**
  Many reviewers, e.g., 9b2j and oWmh, found the early-stopping story unclear.
  We suspect this is related to an oversight in the statement of Theorem 1.1.
  Here we detail this oversight and how we feel it clarifies the early-stopping discussion
  while still giving a bound where all terms may be controlled;
  that said, beyond this technical point, we acknowledge there is a need to revise the body
  to better explain early-stopping, for instance including many points from our individual responses.

    The basic summary is as follows.

    - In the statement of Theorem 1.1, the term $R$
      within $\tau_1$ and within the generalization error term should be
      replaced with roughly
      $R + \sqrt{t \cdot \mathcal{R}(\overline{U}_\infty)}$.
      As such, if
      $\mathcal{R}(\overline{U}_\infty)$ is small, there is no change to our bound,
      whereas if it
      is large, then $\sqrt{t}$ must be neutralized by a large $m$ in $\tau_1$,
      and by a large $n$ in the generalization term.
      Firstly, this means that all terms can still be controlled, and for instance we
      can still prove a consistency result (Corollary 2.3).  We still have our "nice data"
      story, since the new $\sqrt{t}$ term is multiplied by $\sqrt{\mathcal{R}(\overline{U}_\infty)}$.
      Moreover, we feel this new relationship between $t$ and $n$ clarifies our
      early-stopping story.

    - Fixing the proof requires only a small amount of algebra and in particular
      no new machinery or lemmas.
      Concretely, the oversight manifests in the proof of Lemma 2.2, which after all
      includes a test error estimate.
      The proof evaluates two cases to assert that there is an iterate $W_i$ which is
      both close to initialization, and has small training error.  The issue
      is that it is possible that $i$ is much smaller than $t$,
      and training error
      continues to decrease after iteration $i$ but norms increase, and thus
      it is possible that $W_{\leq t}$ selects a late iterate with very large norm.
      The fix is easy and we already have all tools in place for it:
      since the core inequalities in Lemma 2.2 (and the optimization bound from
      Lemma A.9) already have norm terms, we can just expand our argument with a
      worst-case inequality via Cauchy-Schwarz giving the
      norm upper bound on $\\|W_{\leq t} - W_0\\|$
      of roughly $R + \sqrt{t \cdot \mathcal{R}(\overline{U}_\infty)}$.
      This upper bound is moreover natural: if the data is noisy, then the per-iteration risk
      is large, and the per-iteration gradient norm is large, giving a bound that is on the order
      of $\sqrt{t}$ for noisy data (but much smaller for non-noisy data).

    We are eager to discuss this point further, and also early-stopping in general, and note that
    most of our comments were in the individual reviews.

---

### Decision · Program_Chairs · 2021-09-28

**Decision:**

Accept (Spotlight)

**Comment:**

The paper is very well-written, clear, and the reviewers found the main theorem a strong contribution.

Please take into consideration the reviewers comments and suggestions in the final version.

**Consistency Experiment:**

NeurIPS has a long history of experimentation. In 2014, NeurIPS ran an experiment in which 10% of submissions were reviewed by two independent committees to quantify the randomness in the review process. This year, we repeated a variant of this experiment to see how the quality of the review process has changed over time.  This paper was part of the experiment and was therefore assigned to two committees (consisting of reviewers, an Area Chair, and a Senior Area Chair) that reached independent decisions.  If both committees made the same recommendation, this recommendation was followed. If a single committee recommended acceptance, the paper was accepted (with the exception of a few cases in which the other committee identified what we considered a fatal flaw, e.g., an error in a key result).

Both committees reached the same decision: **Accept (Spotlight)**

The other committee assigned to the paper recommended **Accept (Spotlight)**.  You can find the other set of reviews, along with any follow up discussion with the authors here:
https://openreview.net/forum?id=rMKTq-ca0qu